# A functional role of meningeal lymphatics in sex difference of stress susceptibility in mice

Weiping Dai[1,2,3,10], Mengqian Yang[2,3,10], Pei Xia[2,3,10], Chuan Xiao[1,3,4], Shuying Huang[2,3], Zhan Zhang[1,3,4], Xin Cheng [2,3], Wenchang Li[5], Jian Jin[6], Jingyun Zhang[2,3], Binghuo Wu[7], Yingying Zhang[2,3], Pei-hui Wu[5], Yangyang Lin[3,8], Wen Wu[6], Hu Zhao [2,3], Yan Zhang [9] ✉, Wei-Jye Lin [1,3,4] ✉ & Xiaojing Ye [2,3] ✉

Major depressive disorder is one of the most common mental health conditions. Meningeal lymphatics are essential for drainage of molecules in the cerebrospinal fluid to the peripheral immune system. Their potential role in depression-like behaviour has not been investigated. Here, we show in mice, sub-chronic variable stress as a model of depression-like behaviour impairs meningeal lymphatics in females but not in males. Manipulations of meningeal lymphatics regulate the sex difference in the susceptibility to stress-induced depression- and anxiety-like behaviors in mice, as well as alterations of the medial prefrontal cortex and the ventral tegmental area, brain regions critical for emotional regulation. Together, our findings suggest meningeal lymphatic impairment contributes to susceptibility to stress in mice, and that restoration of the meningeal lymphatics might have potential for modulation of depression-like behaviour.

Major depressive disorder (MDD), characterized by persistent depressed mood as well as loss in motivation and pleasure, is a debilitating mental disorder affecting 4.4% of the world's population, and the leading cause of non-fatal health loss[1,2]. Women are more likely than men to suffer from MDD and develop more severe depressive symptoms[3–6]. Despite of the well-documented sex difference in MDD in the clinical reports, the underlying mechanisms remain largely unknown.

Multiple hypotheses have been proposed for the pathological development of depression from different perspectives. At the circuit level, impaired activation of dopaminergic neurons in the ventral tegmental area (VTA) is considered as key to the deficits of the brain reward circuit and lack of motivation in depression[7]. In addition to changes in neurons, abnormalities in neuroglia have also been noticed. For example, in the postmortem brains of MDD patients, decreased expression of astrocytic markers, such as S100β and glial fibrillary acidic protein (GFAP), have been observed in the medial prefrontal cortex (mPFC), a brain region rendering resilience to stress[8–10]. In rodent models, pharmacological clearance of astrocytes or preventing astrocytic release of ATP caused depressive-like phenotypes[11,12].

[1]Brain Research Center, Sun Yat-sen Memorial Hospital and Zhongshan School of Medicine, Sun Yat-sen University, Guangzhou, China. [2]Faculty of Forensic Medicine, Guangdong Province Translational Forensic Medicine Engineering Technology Research Center, Zhongshan School of Medicine, Sun Yat-sen University, Guangzhou, China. [3]Guangdong Province Key Laboratory of Brain Function and Disease, Zhongshan School of Medicine, Sun Yat-sen University, Guangzhou, China. [4]Guangdong Provincial Key Laboratory of Malignant Tumor Epigenetics and Gene Regulation, Guangdong-Hong Kong Joint Laboratory for RNA Medicine, Medical Research Center, Sun Yat-sen Memorial Hospital, Sun Yat-sen University, Guangzhou, China. [5]Department of Joint Surgery, the First Affiliated Hospital, Sun Yat-sen University, Guangzhou, China. [6]Department of Rehabilitation, Zhujiang Hospital, Southern Medical University, Guangzhou, China. [7]Key Laboratory of Stem Cells and Tissue Engineering, Zhongshan School of Medicine, Sun Yat-sen University, Ministry of Education, Guangzhou, China. [8]Department of Rehabilitation Medicine, the Sixth Affiliated Hospital, Sun Yat-sen University, Guangzhou, China. [9]Department of Psychiatry, The Second Xiangya Hospital, Central South University, Changsha, Hunan, China. [10]These authors contributed equally: Weiping Dai, Mengqian Yang, Pei Xia. ✉e-mail: yan.zhang@csu.edu.cn; linwj26@mail.sysu.edu.cn; yexiaoj8@mail.sysu.edu.cn

However, the upstream factors that initiate or sustain these changes in the brain remain unclear.

Repeated stress is an important environmental factor promoting the pathogenesis of depression[13]. It has been noticed for long that repeated stress also induces, in parallel, profound changes in the immune system[14]. Elevated levels of circulating leukocytes and multiple inflammatory cytokines were detected in the peripheral blood samples of MDD patients, and further elevated in treatment-resistant MDD patients[15–19]. Children with higher circulating levels of interleukin-6 are at a greater risk of developing MDD in adulthood[20]. In the reverse fashion, MDD patients have a higher risk of developing inflammatory disorders, such as diabetes and cardiovascular disorders[21–23]. While these findings suggest a strong link between aberrant immune functions with depression, still little is known about how stress changes the immune system to modulate the susceptibility for depression.

The central nervous system (CNS) has long been considered an "immune privilege" organ, lacking the lymphatic vessels to transport immune cells[24]. However, recent studies have confirmed the existence of lymphatic vessels in the dura mater of human and other animals[25–27]. These meningeal lymphatics constantly drain fluid and molecules from the CNS to the periphery by connecting to the deep cervical lymph nodes (dCLNs), as well as playing an important role in the active transportation of immune cells[25,28,29]. Changes in the meningeal lymphatics have been reported to occur and contribute to the disease progression during aging as well as in the neurodegenerative and neurological diseases, including Alzheimer's disease, Parkinson's disease, traumatic brain injury, encephalitis, and brain tumor[29–34]. However, whether stress affects meningeal lymphatics to regulate the development of depression remains to be investigated.

In this work, we use the sub-chronic variable stress (SCVS) paradigm which exposes mice to three alternating stressors across 6 days. This paradigm has been reported to induce depression-like behaviors in females, but not in males, thus providing a suitable model for investigating sex-dependent stress susceptibility[14,35]. Here we report that SCVS impairs meningeal lymphatics in female but not male mice. Gain-of-function and loss-of-function studies further suggest that meningeal lymphatics contributed to sex differences in the stress susceptibility to depression- and anxiety-like behaviors in mice. Our findings therefore reveal the unexpected effects of short-term stress on the morphology of meningeal lymphatics and brain drainage function through the meningeal lymphatics in a sex-dependent manner. Changes in the meningeal lymphatics, in turn, can modulate the stress susceptibility.

## Results

### Sub-chronic variable stress impairs meningeal lymphatics in female but not male mice

To explore the contribution of meningeal lymphatics to sex-different stress susceptibility, we employed the 6-day SCVS paradigm (Supplementary Figs. 1a, 2a). Consistent with previous reports[14,35], after SCVS, mice of both sexes had significant weight loss (Supplementary Figs. 1b, 2b). However, female but not male mice exhibited depression-like behaviors, including decreased grooming in the splash test, increased immobility in the forced swim test and increased latency to feeding in the novelty-suppressed feeding test (Supplementary Figs. 1c–e, 2c–e). Furthermore, the stressed female mice traveled less to the center zone of an open field arena (Supplementary Fig. 1f–i). Thus, SCVS induced both depression- and anxiety-like behaviors in female mice. In contrast, the stressed male mice showed significantly increased traveled distance in the center zone of the open field, indicating an anxiolytic phenotype (Supplementary Fig. 2f–i).

To investigate whether the sex-different behavioral phenotypes were accompanied by changes in meningeal lymphatics, we examined the mRNA levels of *Lyve1*, a marker of lymphatic endothelial cells (LECs)[25,36] in the dura mater, collected after SCVS or from non-stressed control mice. qPCR analysis revealed that SCVS significantly decreased *Lyve1* mRNA level in the dura mater of female but not male mice (Supplementary Fig. 3a–c).

To further validate the impairment of meningeal lymphatics by SCVS in female but not male mice, we injected a 70kD Dextran-Texas Red tracer into the cisterna magna 72 h after the last behavioral test. The mice were sacrificed 1 h later (Fig. 1a–c). We found that in female mice, the intensity of LYVE1 immunofluorescence staining as well as the diameters of LYVE1-labeled meningeal lymphatic vessels in the superior sagittal sinus (SSS), the transverse sinus (TS) and the confluence of sinuses (COS) of the dura mater were significantly reduced by SCVS. There was also a trend toward decreased coverage area of meningeal lymphatic vessels (Fig. 1d–i). In line with the structural deficits of meningeal lymphatics, significantly less intracisternally injected tracer was detected after SCVS in SSS, TS, and COS areas of the dura mater, where the meningeal lymphatics reside (Fig. 1j). The amount of tracer drained to dCLNs was also significantly reduced (Fig. 1p, q). Of note, impairment in the morphology of meningeal lymphatic vessels in female mice could be observed as early as 24 h after the last episode of SCVS (Supplementary Fig. 4).

In contrast, the intensity of LYVE1 immunofluorescence staining, the coverage area as well as the diameters of meningeal lymphatic vessels in the SSS, TS, and COS areas of dura mater remained unchanged after SCVS in male mice (Fig. 1e, k–n). The amount of intracisternally-injected 70kD Dextran-Texas Red tracer detected in SSS, TS, and COS areas of the dura mater as well as in the dCLNs were also unaltered by SCVS in male mice (Fig. 1o, r, s). Although the meningeal lymphatics of female and male mice showed different sensitivity to stress, we found no significant difference in the overall morphology of meningeal lymphatic vessels as well as in the drainage of intracisternally-injected tracer to the dCLNs comparing non-stressed adult female versus male mice (Supplementary Fig. 3d–h). These data suggest that, consistent with previous reports[30,37], the properties of basal adult female and male meningeal lymphatic vessels were similar.

Collectively, these data uncovered that SCVS induced impairment of the meningeal lymphatics in female but not male mice.

### Improvement of meningeal lymphatics alleviates susceptibility to stress in female mice

Could improvement of meningeal lymphatics render female mice resilient to stress? It is well characterized that vascular endothelial growth factor-C (VEGFC), a secreted protein, binds to the vascular endothelial growth factor receptor 3 (VEGFR3), which mainly expresses on LECs, to promote the growth of lymphatic vessels[38]. Previous studies reported that intracisternal injection of adeno-associated virus (AAV) overexpressing VEGFC promoted the growth of meningeal lymphatics and drainage to dCLNs without affecting meningeal blood vessels[30,39]. Of note, although VEGFC has the capacity to signal on other cells in the brain that express VEGFR2 or VEGFR3, intracisternally injected AAV infects cells mostly surrounding meningeal lymphatics in the dura mater, likely due to the direction of the cerebrospinal fluid (CSF) flow, and did not affect meningeal blood vessel coverage or proliferation of neural stem cells in the hippocampus, which could potentially be mediated by VEGFC signaling[30,39]. Thus, this method provides a way to deliver secreted VEGFC toward meningeal lymphatics in a relatively spatially restricted manner. We therefore injected AAV overexpressing VEGFC or enhanced green fluorescent protein (eGFP) as control into the cisterna magna of female mice. One month later, both groups of mice were subjected to 6-day SCVS (Fig. 2a). Consistent with previous reports[30,39], AAV-infected cells labeled by eGFP were detected mostly in the TS, COS, and part of the SSS that covered the olfactory bulb, surrounding LYVE1-labeled meningeal lymphatics (Fig. 2b). We did not observe infected cells in the brain

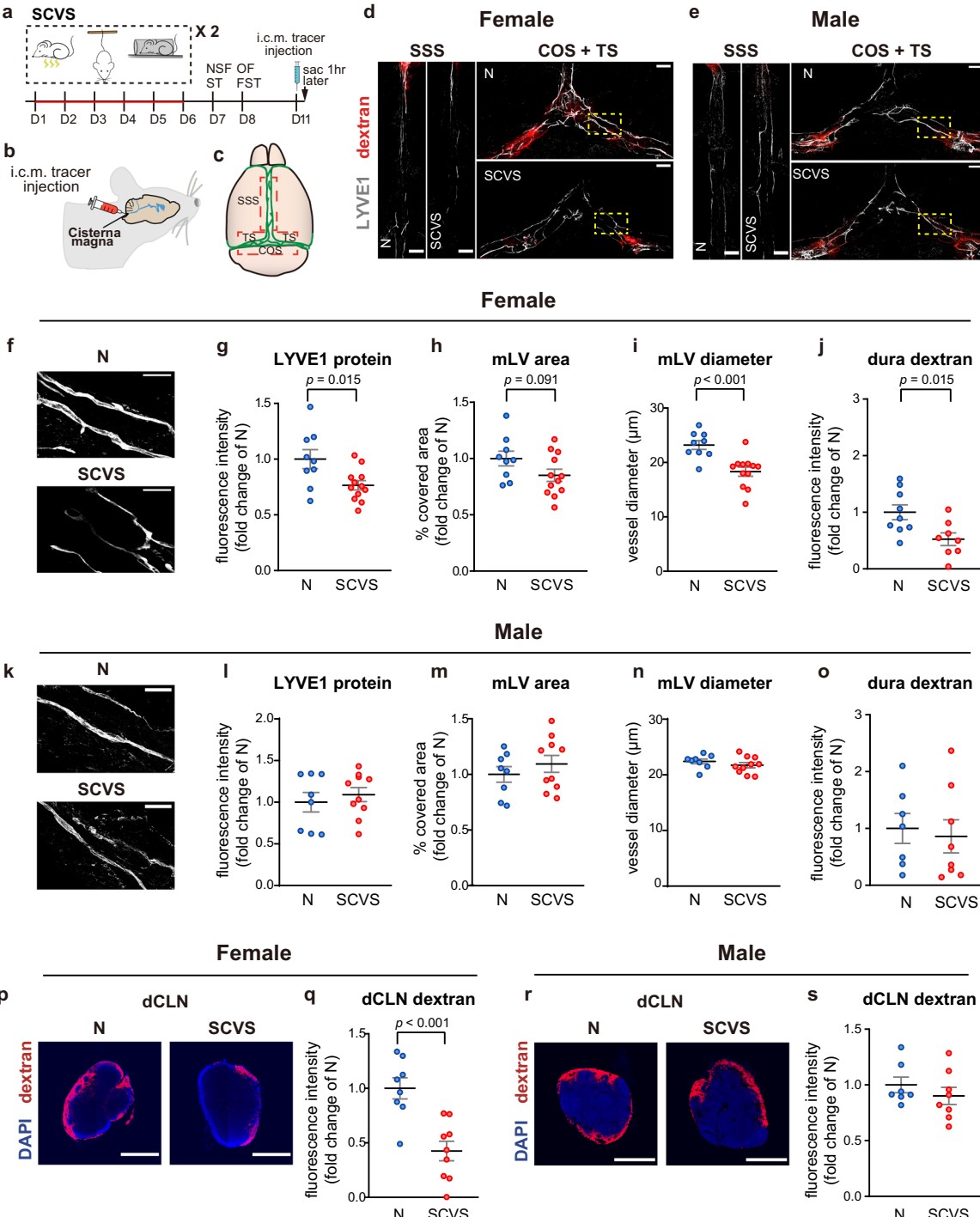

**Fig. 1 | Sub-chronic variable stress (SCVS) impairs meningeal lymphatics in female but not male mice. a** The experimental timeline of the SCVS paradigm, behavioral tests, injection of the 70kD dextran tracer into the cisterna magna (i.c.m.), and tissue collection (sac). **b** Schematic diagram of the i.c.m tracer injection. **c** The schematic diagram of dura mater, with dotted red line bordering the superior sagittal sinus (SSS) as well as the confluence of sinus and transverse sinus (COS + TS) areas chosen for image analyses. **d, e** Representative images depicting the LYVE1 staining (gray) and the dextran tracer (red) in the SSS and COS + TS areas of dura mater of female (**d**) and male (**e**) mice, comparing non-stressed naive group (N) and the SCVS group. Scale bars: 500 μm. **f, k** Representative images depicting LYVE1-labeled meningeal lymphatic vessels (mLV) of female (**f**) and male (**k**) mice at higher magnification for analysis of the diameters. Scale bars: 200 μm. **g–j, l–o** Quantification of the fluorescence intensity of the LYVE1 staining (**g, l**), the

area covered by mLV (**h, m**), and the diameter of LYVE1-labeled mLV (**i, n**) in the SSS and COS + TS areas of dura mater in female (**g–i:** $n = 9$–12 per group; results from three independent experiments) and male (**l–n:** $n = 8$–10 per group; results from two independent experiments) mice. **j, o** Quantification of the fluorescence intensity of the dextran tracer in the SSS and COS + TS areas of dura mater of female (**j:** $n = 8$–9 per group) and male (**o:** $n = 7$-8- per group) mice. **p, r** Representative images depicting the dextran tracer (red) and DAPI (blue) in the deep cervical lymph nodes (dCLN) of female (**p**) and male (**r**) mice. Scale bars: 500 μm. **q, s** Quantification of the fluorescence intensity of dextran tracer in the dCLN of female (**q:** $n = 8$–9 per group) and male (**s:** $n = 7$-8 per group) mice. All data are presented as mean ± s.e.m. and analyzed by unpaired Student's t tests (**g–j, m–o, q, s**) or Mann–Whitney test (**l**). Source data are provided as a Source data file.

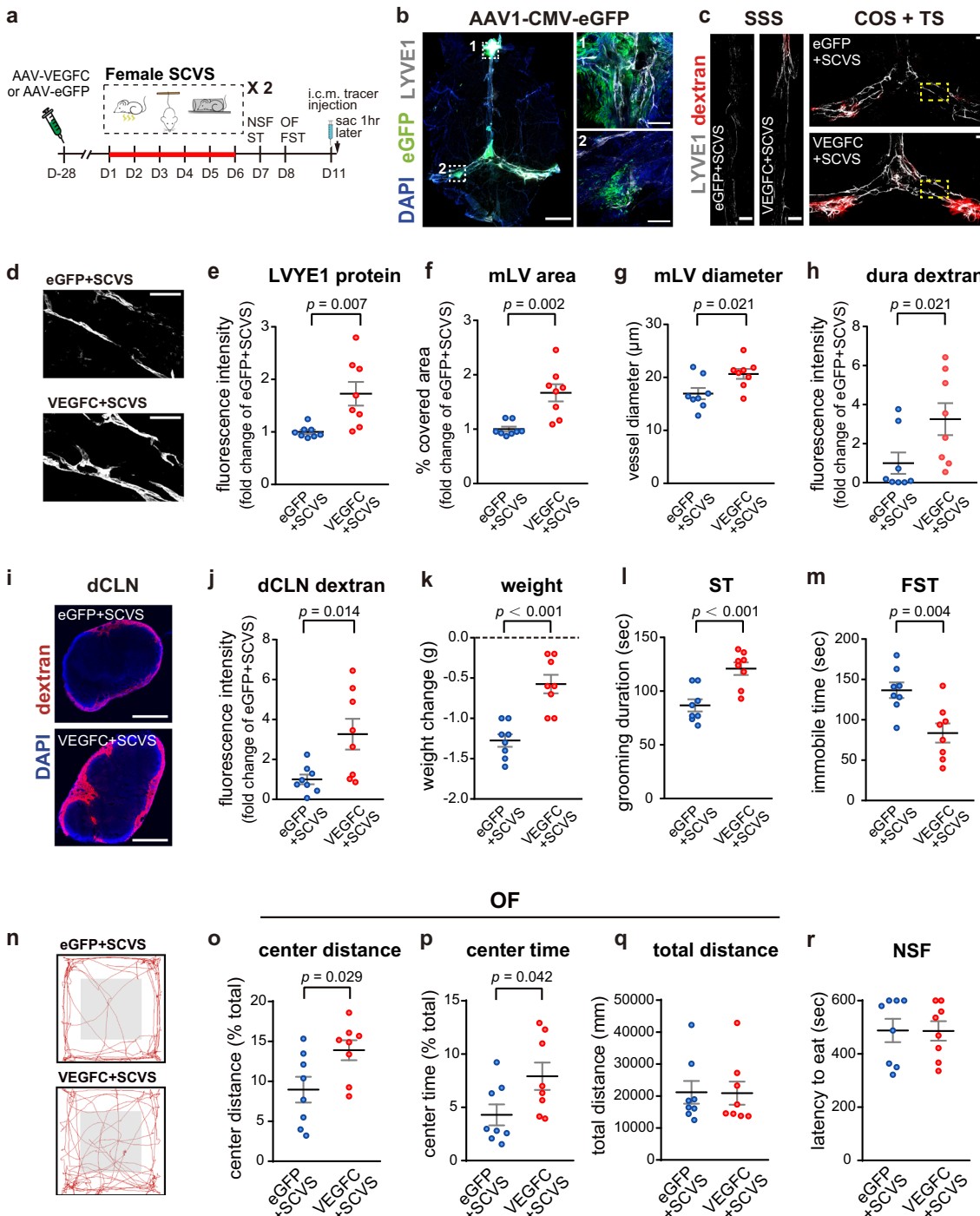

**Fig. 2 | Intracisternal delivery of AAV-VEGFC improves meningeal lymphatics and alleviates stress-induced depression-like behaviors in female mice. a** The experimental timeline of the intracisternal AAV infusion, sub-chronic variable stress (SCVS), behavioral tests, tracer injection into the cisterna magna (i.c.m.), and tissue collection (sac). **b** Representative images depicting eGFP (green)-labeled AAV1-infected cells surrounding LYVE1 (gray)+ meningeal lymphatics (mLV). Nuclei: DAPI (blue). left: whole mount dura mater, scale bars: 2000 μm. right: enlarged views of the boxed areas from the image on the left, scale bars: 200 μm. **c** Representative images depicting the LYVE1 staining (gray) and the dextran tracer (red) in the SSS and COS + TS areas of dura mater, comparing female mice injected with AAV-VEGFC *versus* those injected with AAV-eGFP. Both groups received SCVS. Scale bars: 500 μm. **d** Representative images depicting LYVE1+ mLV at higher magnification. Scale bars: 200 μm. **e–h** Quantification of the fluorescence intensity of the LYVE1 staining (**e**), the area covered by mLV (**f**), the diameter of LYVE1+ mLV (**g**), and

the fluorescence intensity of the dextran tracer in the SSS and COS + TS areas of dura mater (**h**). **i** Representative images depicting the dextran tracer (red) and DAPI (blue) in the deep cervical lymph nodes (dCLN). Scale bars: 500 μm. **j** Quantification of the fluorescence intensity of dextran tracer in the dCLN. **k** Quantification of changes in the body weight. **l** Quantification of grooming duration in the splash test (ST). **m** Quantification of the immobile time in the forced swim test (FST). **n** Representative traces of animal's paths in the open field (OF). **o–q** Quantification of the traveled distance in the center zone as a percentage of the total traveled distance (**o**), the time spent in the center zone as a percentage of the total time (**p**), and total traveled distance (**q**) in the OF. **r** Quantification of the latency to eat in the novelty-suppressed feeding test (NSF) (**e–h, j–m, o–r**: n = 8 per group; results from two independent experiments). All data are presented as mean ± s.e.m. and analyzed by unpaired Student's *t* tests (**e, g, j–m, o, p**) or Mann–Whitney tests (**f, h, q, r**). Source data are provided as a Source data file.

parenchyma, except occasionally few cells in the cerebellum that near needle insertion site. Overexpression of VEGFC significantly enhanced the intensity of LYVE1 immunofluorescence staining in the SSS and TS/COS of dura mater, as well as the diameter and the coverage area of meningeal lymphatics (Fig. 2c–g). Furthermore, these mice showed increased drainage of intracisternally injected tracer into the dura mater and dCLNs (Fig. 2c, h, I, j).

Importantly, the VEGFC-treated female mice had significantly less weight loss after SCVS (Fig. 2k). They also exhibited significantly enhanced grooming duration in the splash test, reduced immobility in the forced swim test, and increased exploration of the center zone in the open field test (Fig. 2l–p). Of note, the latency to feeding in the novelty-suppressed feeding test and general locomotion, as measured by the total distance traveled in the open field test, were not affected (Fig. 2q, r). In the non-stressed female mice, intracisternal injection of AAV-VEGFC promoted meningeal lymphatic growth without affecting the aforementioned behaviors (Supplementary Fig. 5a–m), suggesting that enhancement of meningeal lymphatics per se was not sufficient to alter depression- or anxiety-like behaviors in female mice, but rather, increased their resilience to stress.

Social avoidance and anhedonia are common symptoms of depression[40,41]. Compared to non-stressed group, female mice that experienced SCVS exhibited significant reduction in preferences for social interaction and sucrose water, both of which were prevented by intracisternal injection of AAV-VEGFC (Supplementary Fig. 6a–e).

Taken together, these results suggest that improvement of meningeal lymphatics by intracisternal injection of AAV over-expressing VEGFC in the dura mater prevented stress-induced depression- and anxiety-like behaviors in female mice.

## Improvement of meningeal lymphatics ameliorates stress-induced alterations in the mPFC and the VTA of female mice

Previous studies suggest that meningeal lymphatics are connected to the glymphatic system, a perivascular space aligned by astrocyte end-feet in the brain parenchyma, to facilitate the clearance of molecules from the brain parenchyma to meningeal lymphatics and further to the peripheral lymph nodes[42]. Alteration in meningeal lymphatics could change the rate of cerebrospinal fluid (CSF) influx and interstitial fluid (ISF) diffusion in the brain parenchyma[30,42]. To examine whether CSF influx and diffusion in the brain parenchyma could be affected by SCVS and the VEGFC treatment, we injected ovalbumin (OVA)-Alexa Fluor 647 tracer into the cisterna magna of female mice, and perfused the mice 1 h later. Significantly less tracer was found in the brain parenchyma after SCVS, which was prevented by intracisternal infusion of AAV-VEGFC that improved meningeal lymphatics (Supplementary Fig. 7a–c).

To estimate the amount of intracisternally-injected tracer accumulated in the brain parenchyma and in the ventricles, we collected the whole brains (excluding skulls) from another cohort of female mice and measured the amount of tracer in the tissue homogenates of naive mice and mice subjected to SCVS. No difference was detected between the two groups (Supplementary Fig. 7d–e). Since SCVS reduced tracer detected in the brain parenchyma, in the vicinity of meningeal lymphatics and in the dCLNs (Fig. 1j, p, q and Supplementary Fig. 7a–c), these data suggest that SCVS increased tracer accumulation in the brain ventricles and the subarachnoid space of female mice.

It has been reported that substantial amount of tracer that reached the dura mater could be phagocytosed by macrophages[29,43]. Could SCVS alter the phagocytic activity of macrophages near meningeal lymphatics? Using CX3CR1-eGFP mice to label macrophages, we found that less tracer was detected in the TS of dura mater, along with less tracer detected in the nearby macrophages. However, the tracer in the macrophages as a percentage of total tracer was not significantly altered after SCVS (Supplementary Fig. 7f–k), suggesting that alteration in the phagocytic activity of macrophages was unlikely

to be responsible for the reduction of tracer detected near meningeal lymphatics.

Impairment of CSF perfusion is often associated with deficits in the efflux of ISF macromolecules from the brain parenchyma[30,44]. To examine whether SCVS also impaired efflux of ISF molecules, we injected 70kD and 40kD Dextran tracers into the mPFC and the VTA. For both regions, the clearance of these tracers was significantly reduced by SCVS (Supplementary Fig. 7l–q). Collectively, these data suggest that SCVS led to reduction in CSF perfusion in female mice, which was likely due to impairment of meningeal lymphatics. The decreased CSF perfusion was accompanied by reduction in efflux of ISF macromolecules and reduction of CSF reaching the vicinity of meningeal lymphatics and dCLNs.

Although SCVS and the VEGFC treatment affected the CSF perfusion of the whole brain, we noticed that the distribution of intracisternally-injected tracer in the brain parenchyma was uneven, with an enrichment in the mPFC at 15–30 min post-injection, and an enrichment in the hypothalamus and mid-brain regions including VTA at 30–60 min post-injection, in naive female mice (Fig. 3a, b). Importantly, the mPFC and the VTA are both critically involved in the regulation of emotions[7–10].

The mPFC is considered essential in rendering stress resilience[45]. In addition, previous studies have found hypoactive astrocytes in the mPFC of the postmortem brains of MDD patients[8–10]. Since the glymphatic system is aligned by astrocyte end-feet[42], we reasoned that impairment of drainage through meningeal lymphatics by stress might lead to accumulation of macromolecules in the glymphatic system in the mPFC, which in turn resulted in impairment of mPFC astrocytes. Consistent with the clinical observations[8–10], we found that SCVS significantly reduced the expression of astrocyte markers, S100β and GFAP, as well as the density of S100β-labeled astrocytes in the female mPFC. Improvement of meningeal lymphatics by intracisternal injection of AAV over-expressing VEGFC in the dura mater prevented stress-induced reduction of S100β and GFAP levels as well as astrocyte density in the mPFC (Fig. 3d–h).

In addition, decreased activity of dopaminergic neurons in the VTA has long been implicated in the development of depression[7]. We also found that SCVS reduced the expression of c-FOS, an indicator of neuronal activity[46], in the tyrosine hydroxylase-positive (TH+) dopaminergic neurons in the VTA of female mice. AAV-mediated VEGFC overexpression in the dura mater significantly enhanced c-FOS expression in the TH+ neurons, indicating an improvement in VTA dopaminergic neuronal activity (Fig. 3i–j). Of note, the expression of c-FOS in the mPFC or the expression of GFAP in the VTA was not significantly changed by SCVS (Supplementary Fig. 8). Furthermore, the VEGFC treatment alone did not alter astrocytic protein expression in the mPFC or c-FOS expression in VTA dopaminergic neurons (Supplementary Fig. 5n–t) in non-stressed female mice, suggesting that enhancement of meningeal lymphatics per se was not sufficient to alter the mPFC astrocytes and the VTA dopaminergic neuronal activity.

One of the key physiological functions of lymphatic vessels is to absorb and transport lipids[47]. On the other hand, lipids are enriched and engaged in a wide range of physiological functions in the brain[48]. Previous studies have reported that chronic unpredictable stress alters lipidomic profiles in the prefrontal cortex[49], and the activity of dopaminergic neurons in the VTA can be modulated by triacylglycerols (TAGs)[50]. As efflux of ISF macromolecules decreased after SCVS (Supplementary Fig. 7l–q), we reasoned that impairment of meningeal lymphatics by SCVS might lead to accumulation of lipids in the brain parenchyma, which in turn caused abnormalities in the mPFC and the VTA, whereas the VEGFC treatment prevented these changes. Toward that end, we performed lipidomic analyses of the mPFC, the VTA and the dura mater, comparing three groups of female mice as follows: (1) naive mice injected with AAV-eGFP, (2) mice injected with AAV-eGFP and experienced SCVS, and (3) mice

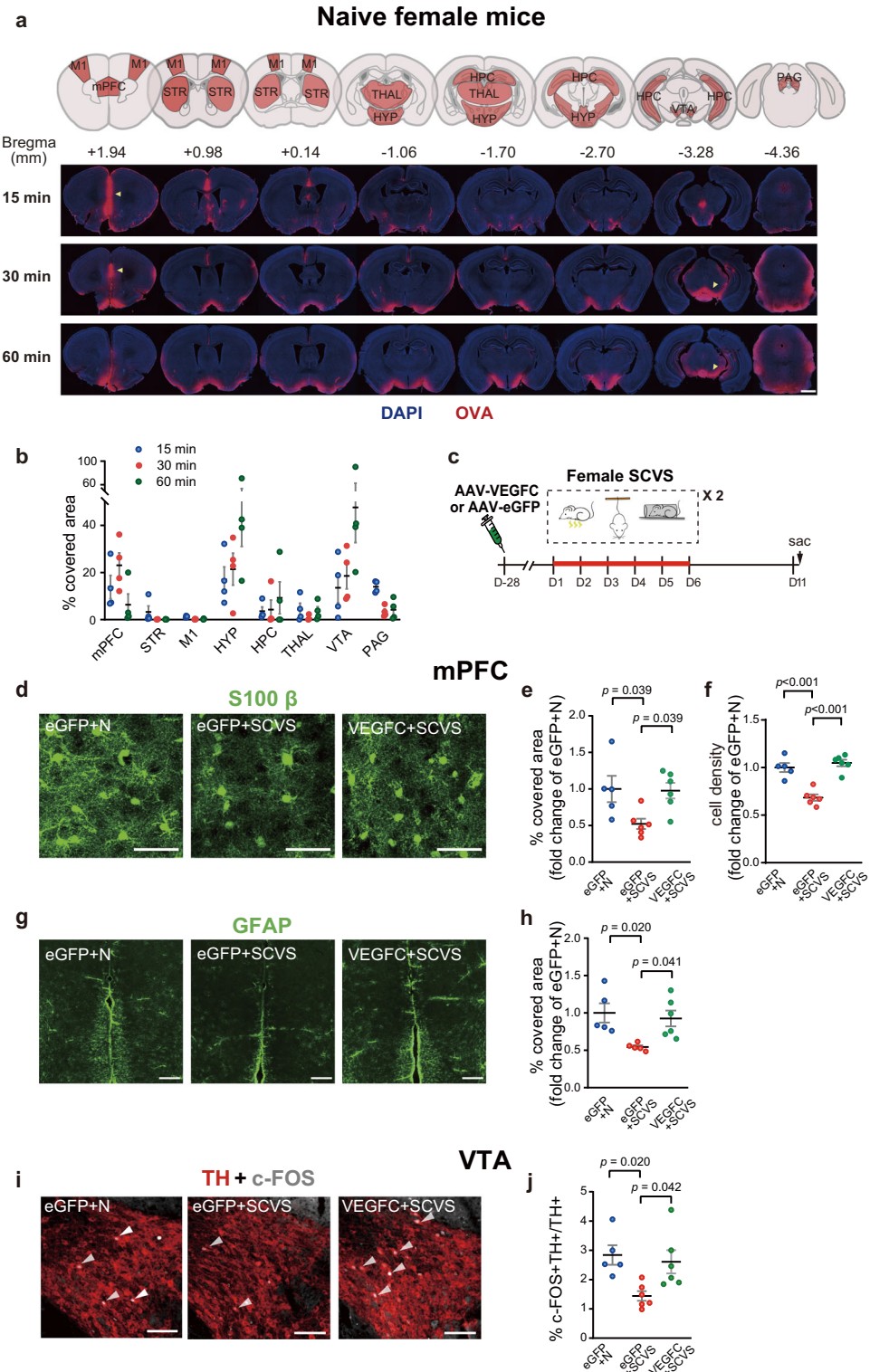

injected with AAV-VEGFC and experienced SCVS (Supplementary Fig. 9a). The partial least squares discrimination analysis (PLS-DA) score plots showed clear separations of the three groups (Supplementary Fig. 9b). After SCVS, significant increase in a variety of lipids including long-chain TAGs was notable in the mPFC while a small subset of TAGs were also increased in the VTA after SCVS, which were partially prevented by the VEGFC treatment (Supplementary Fig. 9c, d). Importantly, the VEGFC treatment significantly increased a panel of lipid molecules, including long-chain TAGs, in the dura mater (Supplementary Fig. 9e). These findings were consistent with the

notion that increased drainage through meningeal lymphatic by the VEGFC treatment might facilitate lipid efflux and reduced stress-induced accumulation of lipid molecules in the mPFC and the VTA of female mice.

To examine the functional link between identified lipids significantly changed by SCVS in the mPFC and the VTA of female mice, we focused on the long-chain TAGs, which were accumulated in both brain regions after SCVS and the accumulation were partially prevented by the VEGFC treatment. The results showed that direct injection of TAGs into the mPFC led to significant loss of astrocytes,

**Fig. 3 | Intracisternal delivery of AAV-VEGFC in female mice rescues sub-chronic variable stress (SCVS)-induced impaired expression of astrocytic markers in the mPFC and cFOS expression in the VTA dopaminergic neurons.** **a** Upper panel, the coronal atlas, with the brain regions of interest highlighted in orange. Lower panel, Representative images of intracisternally-injected OVA-Alexa Fluor 647 tracer (red) distribution in the brain sections at 15, 30, and 60 min post-injection, with DAPI staining (blue). Scale bars: 1000 μm. Yellow arrowheads indicated accumulation of tracer in the mPFC at 15 and 30 min post-injection, and in the VTA at 30 and 60 min post-injection. **b** Quantification of the distribution of tracer in multiple brain regions at different time points post-intracisternal injection ($n = 4$ per group). **c** The experimental timeline of the intracisternal AAV infusion, sub-chronic variable stress (SCVS) paradigm, and tissue collection (sac). **d**, **g** Representative images depicting the S100β (**d**: scale bars: 50 μm) and GFAP (**g**: scale bars: 200 μm) staining in the mPFC. **e**, **h** Quantification of the percentage of covered area by S100β (**e**) and GFAP (**h**) staining in mPFC. **f** Quantification of the density of S100β-labeled astrocytes in the mPFC. **i** Representative images of the TH (red) and c-FOS (gray) staining in the VTA. White arrowheads denote cells dual-labeled by c-FOS and TH. Scale bars: 100 μm. **j** Quantification of the percentage of c-FOS$^+$ neurons in TH-labeled dopaminergic neurons in the VTA (**e**–**j**: $n = 5$–6 per group; results from two independent experiments). All data are presented as mean ± s.e.m. and analyzed by one-way ANOVA followed by Tukey's post hoc tests. Source data are provided as a Source data file.

whereas in the VTA the injection decreased the expression of c-FOS in the TH$^+$ dopaminergic neurons (Supplementary Fig. 9f–i).

Collectively, the findings above suggest that enhancement of meningeal lymphatics by VEGFC prevented SCVS-induced abnormalities in the mPFC astrocytes and the VTA dopaminergic neurons in the female mice, likely through the facilitation of long-chain TAG efflux from the two brain regions.

## Impairment of meningeal lymphatics renders male mice susceptible to stress

Could impairment of meningeal lymphatics in male mice facilitate the induction of depression-like phenotypes by SCVS? The VEGFC binds VEGFR3 to regulate lymphangiogenesis[51]. Intracisternal injection of AAV expressing VEGFR3$_{d1-4}$-IgG Fc has been reported to disrupt the growth of meningeal lymphatic vessels[39]. The VEGFR3$_{d1-4}$-IgG Fc contains the extracellular domains 1–4 of VEGFR3 that bind to VEGFC, but the intracellular domain was replaced with the immunoglobulin fragment crystallizable domain (Fc). Thus, VEGFR3$_{d1-4}$-IgG Fc competes with endogenous VEGFR3 for VEGFC binding but fails to initiate signaling. The control virus expressed the IgG Fc-fused extracellular domains of 4–7 of VEGFR3, which does not bind to VEGFC[39,52–54]. We injected AAV expressing VEGFR3$_{d1-4}$-IgG Fc or VEGFR3$_{d4-7}$-IgG Fc as control into the cisterna magna of male mice (Fig. 4a). The AAV-infected cells were found mostly near LYVE1-labeled meningeal lymphatics in the TS/COS and part of the SSS that covered the olfactory bulb in the dura mater (Fig. 4b). The overexpression of VEGFR3$_{d1-4}$-IgG Fc from the AAV-infected cells would sequester VEGFC in the microenvironment and prevent them from binding to VEGFR3 on the meningeal LECs. Both groups of mice were then subjected to 6-day SCVS (Fig. 4a). The results showed that VEGFR3$_{d1-4}$-IgG Fc overexpression significantly reduced LYVE1 expression in the SSS and TS/COS of dura mater, as well as the diameter and the coverage area of LYVE1-labeled meningeal lymphatics (Fig. 4c–g). In parallel, there were significant decreases in the amount of intracisternally injected tracer detected in the dura mater and dCLNs (Fig. 4c, h–j).

Importantly, the VEGFR3$_{d1-4}$-IgG Fc overexpressed male mice showed significantly decreased grooming time in the splash test, increased immobility time in the forced-swim test, and impaired social interactions after SCVS (Fig. 4k, l and Supplementary Fig. 6f–l). There was no significant difference in the latency to feeding in the novelty-suppressed feeding test, change of body weight, overall locomotion and exploration of the center zone in the open field test, nor in the sucrose preference test between groups of VEGFR3$_{d1-4}$ or VEGFR3$_{d4-7}$ overexpressed male mice (Fig. 4m–r and Supplementary Fig. 6f, j).

Of note, without SCVS, the VEGFR3$_{d1-4}$ treatment alone did not affect the performance of non-stressed male mice in the tests for depression-like behaviors (Supplementary Fig. 10a–d), suggesting that impairment of meningeal lymphatic per se was not sufficient to induce depression-like behaviors in male mice, but increased their susceptibility to stress. There was a trend toward decease in the time spent in the center zone of the open field arena ($p = 0.055$) for the VEGFR3$_{d1-4}$ treatment group, indicating a potential anxiogenic phenotype (Supplementary Fig. 10e–h). Furthermore, intracisternal injection of AAV

overexpressing VEGFC in male mice subjected to SCVS did not affect animals' performances in depression- and anxiety-like behavioral tests, as well as changes in body weight (Supplementary Fig. 10i–q), suggesting existence of a "ceiling effect" for the role of meningeal lymphatics in regulating depression- and anxiety-like behaviors.

Similar to female mice (Fig. 3a, b), there was an enrichment of tracer in the mPFC at 15–30 min post-injection, and in the hypothalamus and the VTA at 30–60 min post-injection in naive male mice (Fig. 5a, b). SCVS did not significantly alter astrocytic protein expression in the mPFC, nor alter c-FOS expression in the TH$^+$ dopaminergic neurons in the VTA of male mice. However, the VEGFR3$_{d1-4}$ treated male mice exhibited significantly less S100β and GFAP expression as well as reduced density of S100β$^+$ astrocytes in the mPFC, and less c-FOS expression in the TH$^+$-dopaminergic neurons in the VTA after SCVS (Fig. 5c–j).

Due to the concern of potential lymphatic-independent effects of manipulation of VEGFC-VEGFR3 signaling on stress susceptibility, we also examined whether ligation of the lymphatic afferent vessels to the dCLNs, which blocks the drainage pathway through meningeal lymphatics[28,29], rendered male mice susceptible to SCVS. Compared to sham-operated mice, male mice received the ligation surgery showed less intracisternally-injected tracer drained to the dCLNs (Fig. 6a–c). Moreover, these mice exhibited decreased grooming duration in the splash test and increased immobility in the forced-swim tests after SCVS, without significant alterations of behaviors in the novelty-suppressed feeding test and the open field test as well as body weight changes (Fig. 6d–k). These mice also showed decreased S100β and GFAP expression as well as reduced density of S100β$^+$ astrocytes in the mPFC, and reduced c-FOS expression in the TH$^+$ dopaminergic neurons in the VTA after SCVS (Fig. 6l–r).

Collectively, the data above revealed that impairment of meningeal lymphatics or drainage through meningeal lymphatics increased the susceptibility of male mice to stress, and facilitated stress-induced abnormalities in the mPFC and the VTA.

## Sub-chronic variable stress changes the transcriptional profile of meningeal lymphatic endothelial cells in female mice

To explore the mechanism by which stress regulates meningeal lymphatics, we wondered whether the differential impairment of meningeal lymphatics in female but not male mice by SCVS was due to sex difference in the level of corticosterone, the stress hormone considered as important for pathological development of depression and anxiety[55,56]. Toward that end, we measured serum corticosterone levels before and after the completion of SCVS in both female and male mice (Supplementary Fig. 11a). Consistent with previous reports on circadian oscillation of corticosterone levels[57], the serum corticosterone levels were higher at 8:00 PM as compared with those at 8:00 AM in the naive female and male mice without detectable sex difference. Notably, when comparing serum corticosterone levels at 8:00 AM, increases in corticosterone levels were evident in both female and male mice after SCVS as compared to the naive mice. However, dramatic decreases in corticosterone levels were observed in female and male mice after SCVS when the serum corticosterone levels were

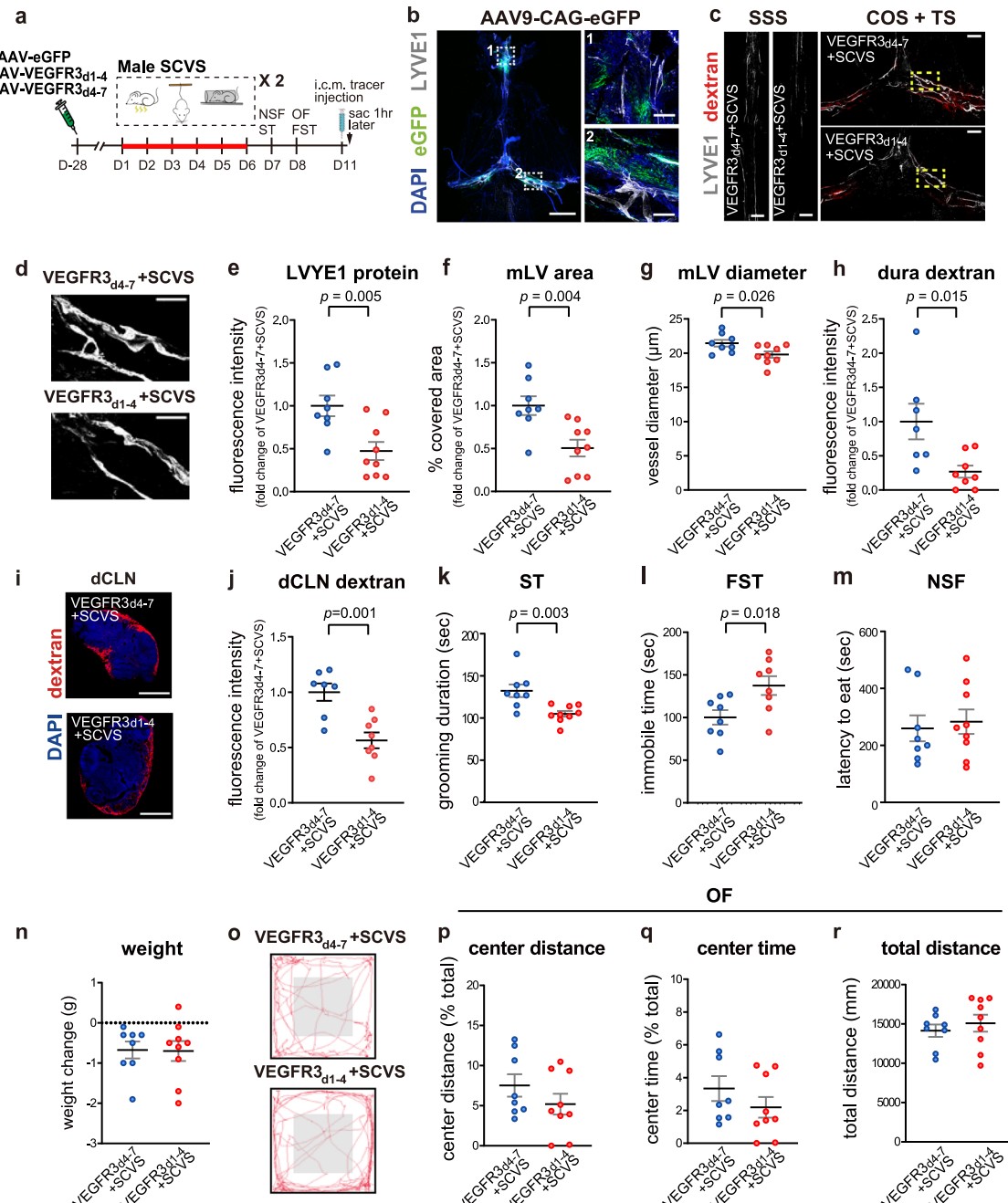

**Fig. 4 | Intracisternal delivery of AAV-VEGFR3$_{d1-4}$ impairs meningeal lymphatics and promotes stress-induced depression-like behaviors in male mice. a** The experimental timeline of the AAV infusion, sub-chronic variable stress (SCVS), behavioral tests, trace injection into the cisterna magna (i.c.m.), and tissue collection (sac). **b** Representative images of eGFP (green)-labeled AAV9-infected cells surrounding LYVE1 (gray)-positive meningeal lymphatics (mLV). Nuclei: DAPI (blue). Left: whole mount dura mater, scale bars: 2000 μm. Right: enlarged views of the boxed areas from the image on the left, scale bars: 200 μm. **c** Representative images of the LYVE1 staining (gray) and the dextran tracer (red) in the SSS and COS + TS areas of dura mater, comparing male mice injected with AAV-VEGFR3$_{d1-4}$ (VEGFR3$_{d1-4}$ + SCVS) with those injected with AAV-VEGFR3$_{d4-7}$ (VEGFR3$_{d4-7}$ + SCVS). Both groups experienced SCVS. Scale bars: 500 μm. **d** Representative images of LYVE1$^+$ mLV at higher magnification. Scale bars: 200 μm. **e–h** Quantification of the fluorescence intensity of the LYVE1 staining (**e**), the area covered by mLV (**f**), the diameter of LYVE1$^+$ mLV (**g**), and the fluorescence intensity of the dextran tracer in the SSS and COS + TS areas of dura mater (**h**). **i** Representative images of dextran tracer (red) and DAPI (blue) in the dCLNs. Scale bars: 500 μm. **j** Quantification of the fluorescence intensity of dextran tracer in the dCLNs. **k–n** Quantification of grooming duration in the splash test (ST, **k**), the immobile time in the forced swim test (FST, **l**), the latency to eat in the novelty-suppressed feeding test (NSF, **m**) and changes in the body weight (**n**). **o** Representative traces of animal's paths in the open field (OF). The center zone: the gray box in the center. **p–r** Quantification of the traveled distance in the center zone as a percentage of the total traveled distance (**p**), the time spent in the center zone as a percentage of the total time (**q**), and total traveled distance (**r**) in the OF (**e–g**, **k–n**, **p–r**: $n = 8$–9 per group; **h**, **j**: $n = 7$–8 per group; results from two independent experiments). All data are presented as mean ± s.e.m. and analyzed by unpaired Student's $t$ tests (**e–h**, **j–l**, **n**, **p–r**) or Mann–Whitney test (**m**). Source data are provided as a Source data file.

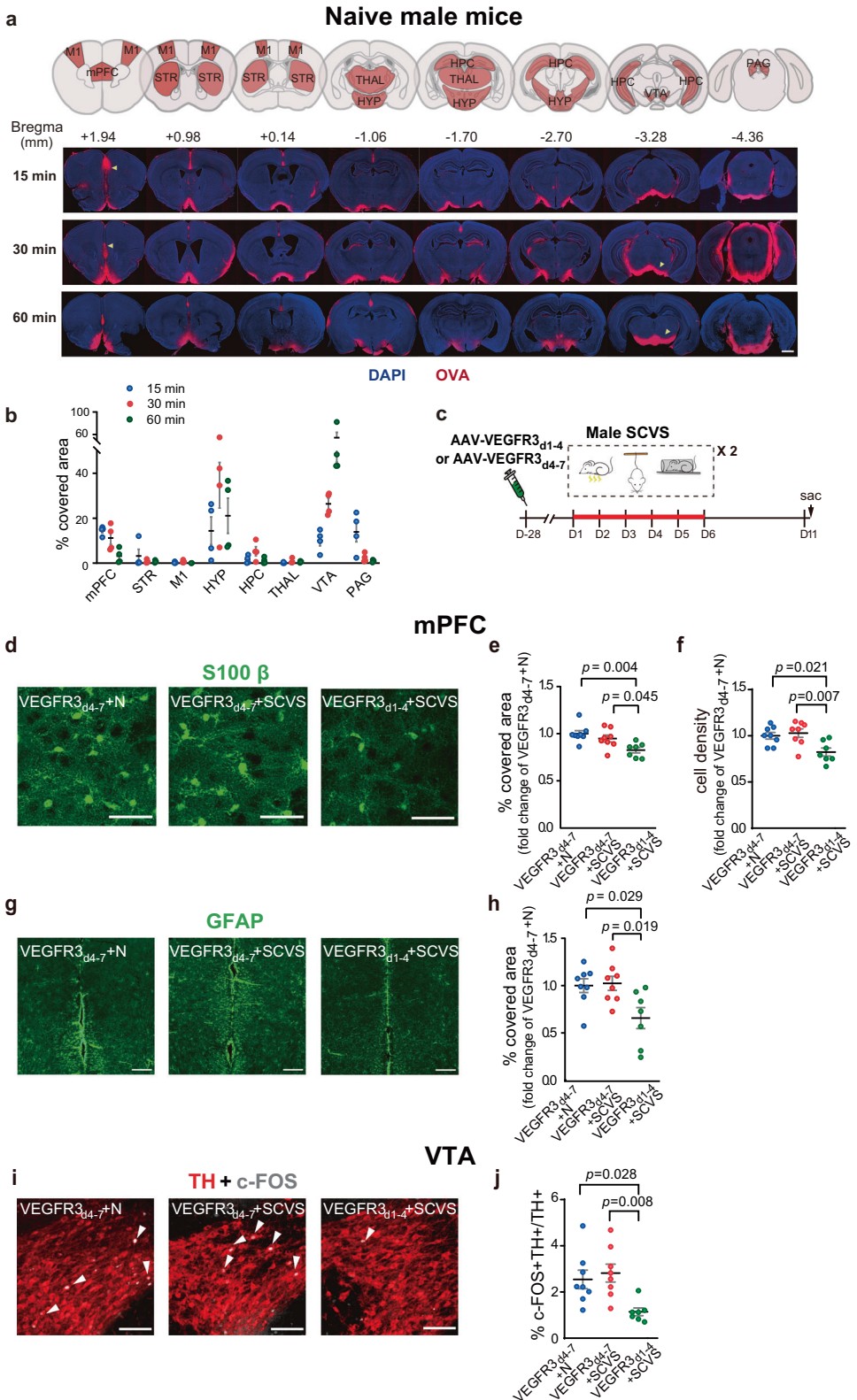

examined at 8:00 PM (Supplementary Fig. 11b, c). Therefore, the circadian oscillation of serum corticosterone levels was blunted after SCVS, with similar magnitudes of changes detected in both female and male mice. To directly examine the effect of corticosterone on meningeal lymphatics, male mice received consecutive corticosterone injection for 28 days, which generated depression-like phenotypes as reported previously (Supplemental Fig. 11d–f)[58]. However, the chronic corticosterone administration did not induce detectable changes in the meningeal lymphatics as measured by the LYVE1 expression, the coverage of LYVE1-labeled meningeal lymphatics as well as the diameter of meningeal lymphatics (Supplemental Fig. 11g–k). Our findings therefore suggested that corticosterone was unlikely to be responsible for the SCVS-induced sex-dependent impairment of meningeal lymphatics.

**Fig. 5 | Intracisternal delivery of AAV-VEGFR3$_{d1-4}$ in male mice promotes sub-chronic variable stress (SCVS)-induced impaired expression of astrocytic markers in the mPFC and cFOS expression in the VTA dopaminergic neurons. a** Upper panel, the coronal atlas, with the brain regions of interest highlighted in orange. Lower panel, Representative images of intracisternally-injected OVA-Alexa Fluor 647 tracer (red) distribution in the brain sections at 15, 30, and 60 min post-injection, with DAPI staining (blue). Scale bars: 1000 μm. Yellow arrowheads indicated accumulation of tracer in the mPFC at 15 and 30 min post-injection, and in the VTA at 30 and 60 min post-injection. **b** Quantification of the distribution of tracer in multiple brain regions at different time points post-intracisternal injection (*n* = 4 per group). **c** The experimental timeline of the intracisternal AAV infusion, SCVS

paradigm and tissue collection (sac). **d, g** Representative images depicting the S100β (**d**: scale bars: 50 μm) and GFAP (**g**: scale bars: 200 μm) staining in the mPFC. **e, h** Quantification of the percentage of covered area by S100β (**e**) and GFAP (**h**) staining in mPFC. **f** Quantification of the density of S100β-labeled astrocytes in the mPFC. **i** Representative images of the TH (red) and c-FOS (gray) staining in the VTA. White arrowheads denote cells dual-labeled by c-FOS and TH. Scale bars: 100 μm. **j** Quantification of the percentage of c-FOS⁺ neurons in TH-labeled dopaminergic neurons in the VTA (**e, f, h, j**: *n* = 7−8 per group; results from two independent experiments). All data are presented as mean ± s.e.m. and analyzed by one-way ANOVA followed by Tukey's post hoc tests. Source data are provided as a Source data file.

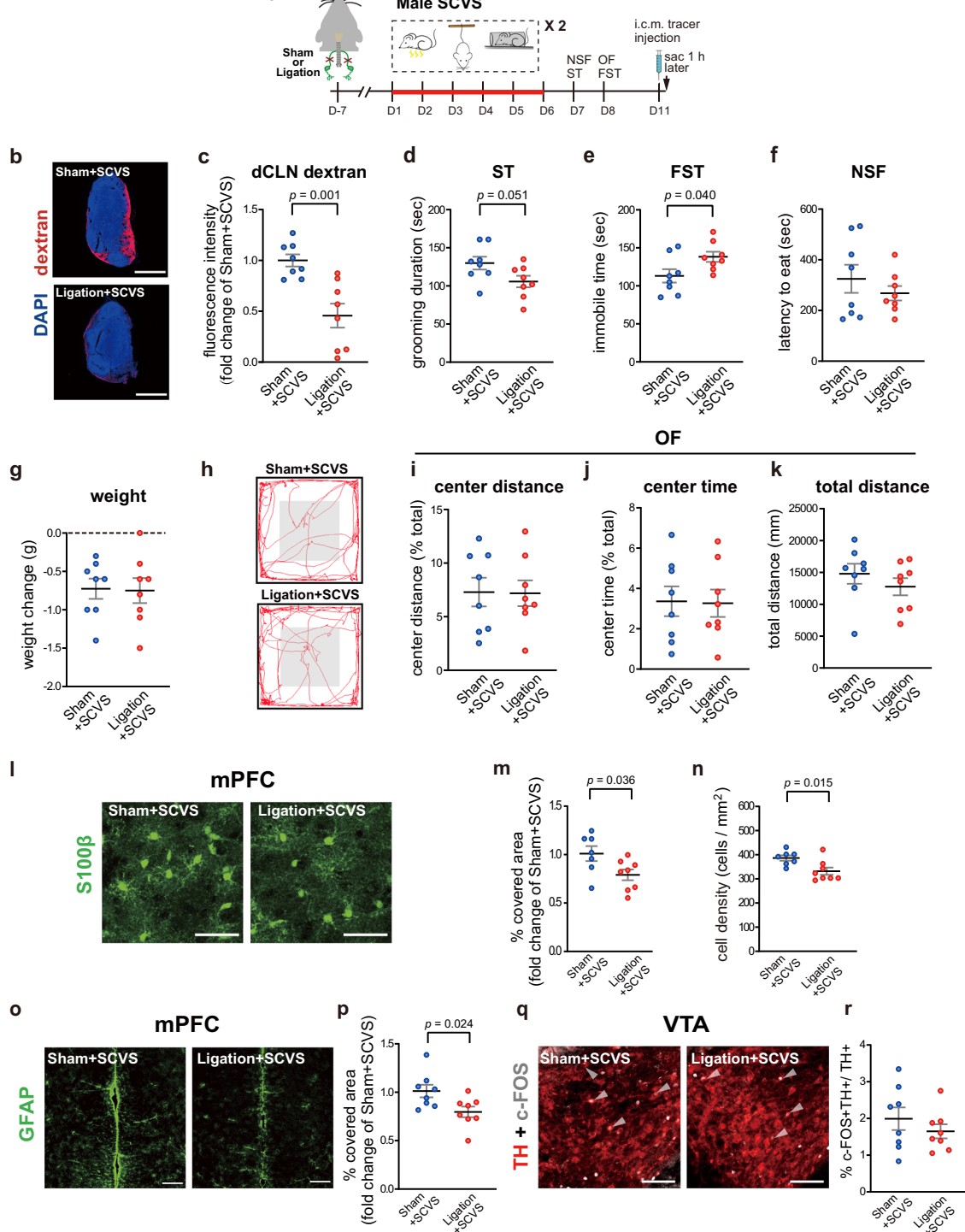

**Fig. 6 | Surgical ligation of the lymphatic vessels afferent to the deep cervical lymph nodes (dCLN) in male mice promotes sub-chronic variable stress (SCVS)-induced depression-like behaviors as well as reduction of astrocytic protein expression in the mPFC and c-FOS expression in the VTA dopaminergic neurons. a** The experimental timeline of the surgical ligation of the lymphatic vessels afferent to the dCLN, sub-chronic variable stress (SCVS) paradigm, behavioral tests, injection of the dextran tracer into the cisterna magna (i.c.m.), and tissue collection (sac). **b** Representative images depicting the dextran tracer (red) and DAPI (blue) in the dCLN. Scale bars: 500 μm. **c** Quantification of the fluorescence intensity of dextran tracer in the dCLN. **d** Quantification of grooming duration in the splash test (ST). **e** Quantification of the immobile time in the forced swim test (FST). **f** Quantification of the latency to eat in the novelty-suppressed feeding test (NSF). **g** Quantification of changes in the body weight by SCVS. **h** Representative traces of animal's paths in the open field (OF). **i**–**k** Quantification of the traveled distance in the center zone as a percentage of the total traveled distance (**i**), the time spent in the center zone as a percentage of the total time (**j**), and total traveled distance (**k**) in the OF. **l**, **o** Representative images of the S100β (**l**: scale bars: 50 μm) and GFAP (**o**: scale bars: 200 μm) staining in the mPFC. **m**, **p** Quantification of the percentage of covered area by S100β (**m**) and GFAP (**p**) staining in mPFC. **n** Quantification of the density of S100β-labeled astrocytes in mPFC. **q** Representative images of the TH (red) and c-FOS (gray) staining in the VTA. White arrowheads denote cells dual-labeled by c-FOS and TH. Scale bars: 100 μm. **r** Quantification of the percentage of c-FOS⁺ neurons in TH-labeled dopaminergic neurons in the VTA (**c**–**g**, **i**–**k**, **p**, **r**: $n = 8$ per group; **m**, **n**: $n = 7$–8 per group; results from two independent experiments). All data are presented as mean ± s.e.m. and analyzed by unpaired Student's $t$ tests. Source data are provided as a Source data file.

To further explore the molecular mechanisms by which stress affects meningeal lymphatics, we performed RNA-seq analysis of meningeal LECs (CD45⁻CD31⁺PDPN⁺) sorted from female mice subjected to SCVS and from non-stressed counterparts (Fig. 7a, b). We found that the percentage of LECs in the total viable cells were significantly lowered in the dura mater of stressed female mice compared to non-stressed mice, further supporting our observation above that SCVS impaired meningeal lymphatics (Fig. 7c). No significant changes were detected in other meningeal cell types, including the blood endothelial cells (CD45⁻CD31⁺PDPN⁻), leukocytes (CD45⁺) or other stromal cells (CD45⁻CD31⁻, composed mostly of fibroblast-like cells) (Fig. 7d–f)[43].

Deseq2 analysis with a cutoff of $p$-adjusted <0.1 identified a total of 236 differentially expressed genes (DEGs), with 156 upregulated and 80 downregulated by SCVS in the meningeal LECs (Fig. 7g, h). Notably, the expression of epidermal growth factor receptor (*Egfr*) was among the top downregulated genes by stress, indicating a deficit in signaling supporting lymphatic sprouting[59] (Fig. 7h). We further performed gene set enrichment analysis (GSEA), which revealed that gene sets involved in the regulation of DNA replication, sphingolipid biosynthesis, serotonin uptake were enriched in the meningeal LECs harvested from the stressed female mice, whereas gene sets engaged by proteasome assembly, ribosome disassembly, platelet-derived growth factor binding were de-enriched, suggesting that stress-induced profound functional changes of meningeal lymphatics (Fig. 7i). It is worth noting that the enrichment in the gene set of serotonin uptake (normalized enrichment score = 2.14, false discover rate = 0.059) in the meningeal LECs of stressed female mice indicates possible involvement of serotonin, a neurotransmitter critically engaged in depression, in mediating stress-induced impairment of meningeal lymphatics (Fig. 7i, j)[60,61].

Previous studies have reported impairment of meningeal lymphatics by aging[30,32]. In addition, stress can accelerate brain aging and the progress of aging-related neurodegenerative diseases[62,63]. Thus, we wondered whether the regulation of meningeal LECs by stress and aging shared any common mechanism. Toward that end, we compared the DEGs identified in the current study with those identified in a previous study examining how aging affected the transcriptome of meningeal LECs[30]. Interestingly, we found that several genes encoding collagen, including *Col6a1*, *Col1a2*, and *Col3a1* as well as a gene encoding a matrix metalloproteinase that processes collagen, *Mmp23*, were significantly downregulated by both stress and aging, suggesting that dissociation from the extracellular matrix might be a key event mediating meningeal lymphatic degeneration (Fig. 7k). To further explore whether gene expression changes identified in the stressed female mice in the current study share any similarity with those associated with human depression, we compared the DEGs identified in the current study with those identified in a previous study by Labonté et al.[64] which explored changes in transcriptional profiles across 6 brain regions from postmortem tissues of patients suffered from female MDD. This analysis identified collectively 12 upregulated and 9 downregulated DEGs shared by LECs of female mice subjected to SCVS with at least one of the examined brain regions by Labonté et al. Among these DEGs, the most noticeable one is Trpm1, which encodes a transient receptor potential cation channel, and was upregulated in the LEC of female mice subjected to SCVS in our study as well as in ventral subiculum, dorsolateral prefrontal cortex, nucleus accumbens and anterior insula of female MDD patients, indicating it as a candidate gene worthy of further exploration for the development of depression. We also noticed that Col1a2, a downregulated gene in the LEC shared by aging and stress, was also downregulated in the nucleus accumbens of female MDD patients (Fig. 7l).

From the RNA-seq data, we noticed a robust induction of chemokine (C-C motif) ligand 6 (*Ccl6*) in the female meningeal LECs by SCVS (Fig. 7h). To examine the potential regulatory role of CCL6 in SCVS-induced meningeal lymphatic impairment in female mice, we intracisternally infused CCL6-neutralizing antibody to disrupt CCL6 signaling[65]. The results showed that functional blockage of CCL6 prevented SCVS-induced depression-like phenotypes and impairment of meningeal lymphatics in female mice (Fig. 8). These findings suggest upregulation of CCL6 as a potential mechanism underlying reduction of meningeal lymphatic coverage induced by SCVS in the female mice. Of note, from the RNA-seq results, the expression of C-C motif chemokine receptor 1 (*Ccr1*, the receptor for CCL6) was low in the meningeal LECs of female mice in the naive group (average counts per million reads: 0.662). *Ccr1* mRNA levels were elevated in 2 out of the 5 biological replicates in the SCVS group, but remained low in the other 3 replicates (average counts per million reads for the SCVS group: 19.660; adjusted $p = 0.999$ compared to the naive group). Based on these data, we suspect that CCL6 is likely acting in cell non-autonomous manner, by signaling on dural cellular intermediaries other than LECs, which in turn leads to reduction in meningeal lymphatic coverage of female mice after SCVS.

## Discussion

In summary, our data showed that SCVS impairs the structure of meningeal lymphatics as well as brain drainage function in the female but not male mice, which is accompanied by profound transcriptomic changes of meningeal LECs. Improvement of meningeal lymphatics by intracisternal delivery of AAV overexpressing VEGFC alleviates stress-induced depression- and anxiety-like behaviors, as well as alterations in the mPFC and VTA in female mice, whereas impairment of meningeal lymphatics by intracisternal injection of AAV overexpressing VEGFR3$_{d1-4}$ or reduction of drainage through meningeal lymphatics by ligation of afferent vessels to the dCLNs increases the susceptibility of male mice to SCVS. Collectively, these findings reveal a functional role of meningeal lymphatics in mediating sex difference in susceptibility to stress.

Since the (re)discovery of meningeal lymphatics in 2015, emerging evidences have suggested their involvement in a variety of neurological disorders[25,28–34,66–70]. Encephalomyelitis, brain tumors, and

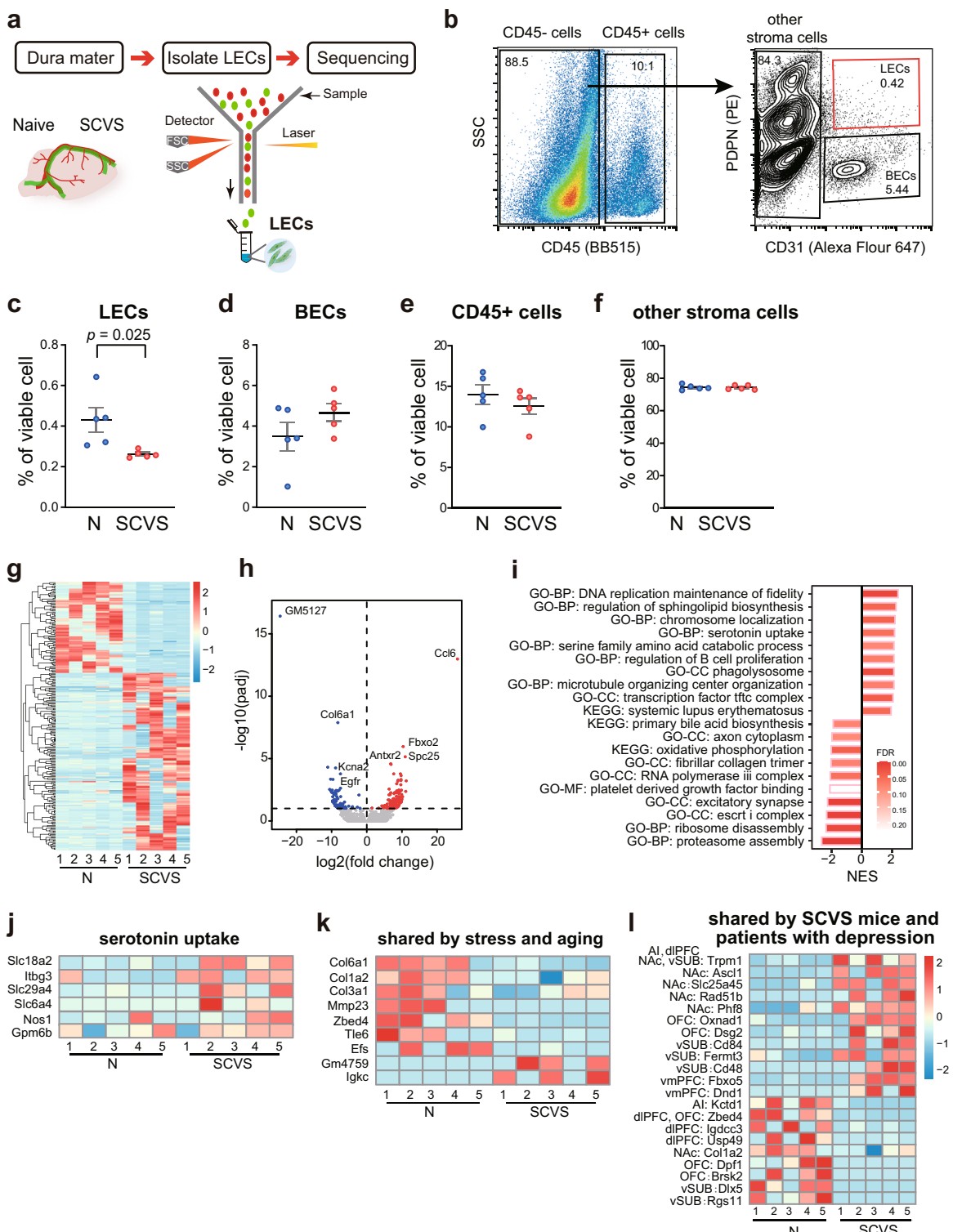

brain injury have been reported to stimulate meningeal lymphangiogenesis, whereas aging and neurodegenerative diseases impairs meningeal lymphatics[29–34,66–70]. However, such changes often develop over a relatively long period of time and/or accompanied by dramatic brain damage. For example, the structural impairment of meningeal lymphatics in the 5xFAD mouse model of Alzheimer's disease were not observed until the age of 13–14 months, which represents a late stage of the disorder[30,67]. Here, we show that as short as 6 days, 1 h per day of stress can impair meningeal lymphatics, suggesting that the adult meningeal lymphatics are readily undergoing plastic changes upon

even relatively mild stimulation. These findings have advanced the current understanding of the regulation of meningeal lymphatics. Furthermore, our data suggest that the reduced coverage and diameter of meningeal lymphatics by SCVS in female mice likely alters the CSF drainage, resulting in reduction of CSF influx and ISF efflux in the brain parenchyma, and reduction of CSF reaching the vicinity of meningeal lymphatics. Such alteration in the CSF drainage might be caused by impairment in the drainage capacity of meningeal lymphatics due to its morphological changes. However, we cannot exclude the possibility that alterations in the pathway that connects the brain

**Fig. 7 | Sub-chronic variable stress (SCVS) induces transcriptional changes of meningeal lymphatic endothelial cells in female mice. a** Schematic of harvesting the dura mater, isolating meningeal lymphatic endothelial cells (LECs) by fluorescence-activated cell sorting (FACS) and RNA sequencing. **b** Representative dot and contour plots showing the gating strategy used to isolate meningeal LECs (CD45⁻PDPN⁺CD31⁺). **c–f** Quantification of the percentages of LECs (**c**), blood endothelial cells (BECs, gated as CD45⁻PDPN⁻CD31⁺) (**d**), CD45⁺ leukocytes (**e**) and other stromal cells (CD45⁻CD31⁻) (**f**) in viable cells, comparing female SCVS mice with non-stressed naive mice (N) (5 biological replicates per group. Each replicate was pooled from 3 to 4 animals). Data are presented as mean ± s.e.m. and analyzed by unpaired Student's *t* tests. **g** Heatmap showing relative expression levels of differentially expressed genes (DEGs, adjusted *p* < 0.1 by Deseq2) in the meningeal LECs, comparing naive (N) with SCVS groups. Color scale bar values represent standardized rlog-transformed values across samples. **h** Volcano plot showing gene expression changes by comparing naive (N) with SCVS groups. Blue and red dots represent significantly downregulated and upregulated genes, respectively. **i** Top enriched (normalized enrichment score, NES > 0; false-discovery rate, FDR < 0.25) and de-enriched (NES < 0, FDR < 0.25) gene sets in the SCVS samples revealed by gene set enrichment analysis (GSEA). GO-BP: gene ontology – biological process, GO-CC: gene ontology – cellular component, GO-MF: gene ontology – molecular function, KEGG: the Kyoto Encyclopedia of Genes and Genomes signaling pathway. **j** Heatmap of DEGs enriched in the serotonin uptake pathway. **k** Heatmap of DEGs co-regulated by stress and aging. **l** Heatmap of DEGs shared by LECs of female mice after SCVS and brain regions of depressed humans. AI anterior insula, NAc nucleus accumbens, vSUB ventral subiculum, dlPFC dorsolateral PFC, vmPFC ventromedial PFC, OFC orbitofrontal cortex. Source data are provided as a Source data file.

perivascular/subarachnoid spaces and the dual stroma occur. Of note, tracer that reaches the dura mater can be phagocytosed by macrophages[29,43]. In our study, the tracer detected in the dura macrophages as a percentage of total tracer was not significantly altered after SCVS, indicating that changes in macrophage activity is unlikely involved in the reduction of tracer detected near meningeal lymphatics.

We acknowledge that, although the SCVS paradigm is a commonly used model for studying sex differences in stress-induced depressive-like behaviors in rodents, our findings should be interpreted carefully since the 6 days stress paradigm may not well-translate into the nature of human depression and other types of animal models for depression that are induced by prolonged chronic stress. We are also aware of the limitation of our current findings, which only showed the impairment of meningeal lymphatics in the SCVS female mice that was detected at 24 h and 5 days after the completion of SCVS paradigm. Whether the state of meningeal lymphatic may continue at a later time point of female mice that experienced SCVS, and its mechanistic link with depressive-like behaviors in other chronic stress models, remain to be determined.

What are the mechanisms underlying stress-induced impairment of meningeal lymphatics? Although corticosterone is a well-recognized stress hormone important for the pathological development of depression and anxiety[55,56], we did not detect sex differences in serum corticosterone levels at baseline and after SCVS. Furthermore, chronic administration of corticosterone did not induce detectable changes in the meningeal lymphatics. Therefore, corticosterone was unlikely to contribute to the sex difference in SCVS-induced meningeal lymphatic impairment. From the RNA-seq analysis of meningeal LECs and functional manipulation experiments, our study suggests SCVS-induced upregulation of CCL6, a chemokine previously reported to regulate the pathogenesis of IL-13-induced tissue inflammation and the homeostasis of hematopoietic stem cells[65,71], as a potential mechanism underlying reduction of meningeal lymphatic coverage in female mice. It is also worth noting that the CCL6 ortholog gene product in human, CCL23, has been reported as a diagnostic biomarker in the serum or cerebrospinal fluid of patients with Alzheimer's disease and ischemic stroke[72,73]. Whether human CCL23 may be involved in the regulation of meningeal lymphatics and the disease progression of neurodegenerative disorders is warranted for further studies.

Interestingly, our RNA-seq analysis of meningeal LECs also revealed that the expression of collagens and their modifier, which were previously found to be decreased by aging[30], were also reduced by stress. Lymphatic vessels are attached to the extracellular matrix composed of collagens and other structural proteins, which supports their development and maintenance[74–79]. Softening of the extracellular matrix reduces proliferation of endothelial cells[74,76,77,79]. Therefore, we speculate that the reduction in collagens and their modifier by stress and aging may lead to matrix softening which in turn impair meningeal lymphatics. In addition, our RNA-seq analysis also identified changes in the enrichment of a number of gene sets, whose contributions to stress-induced deficits of meningeal lymphatics await further examination. Of note, the gene set of serotonin uptake was among the top enriched terms. Serotonin is considered essential for reward processing, and that selective serotonin reuptake inhibitors are standard treatments for depression[60,61,80,81]. It will be of interest to examine in the future studies whether serotonin signaling contributes to stress-induced impairment of meningeal lymphatics and how the upregulation of genes for serotonin uptake process in the meningeal LECs by stress affects the development of depression.

Importantly, our data suggest that changes in the meningeal lymphatics modulate the sex difference in susceptibility to stress. Gain- and loss-of-function manipulations of meningeal lymphatics prevented and facilitated, respectively, SCVS-induced depression- and anxiety-like behaviors. We further show that manipulation of meningeal lymphatics led to changes in the mPFC and the VTA, two key brain regions for emotional regulation[7–10]. Such changes might underlie alterations in susceptibility to stress. We found these data surprising, however, because the meningeal lymphatics are not in direct contact with the neural circuits for emotion. Previous studies have suggested that macromolecules in the interstitial fluid (ISF) and cerebrospinal fluid (CSF) of brain parenchyma can be drained to the meningeal lymphatics and further to the dCLNs[25,28,29,42]. In mouse model of Alzheimer's disease, blockade of this drainage system enhanced accumulation of amyloid-beta in the brain parenchyma and altered hippocampal gene expression[30]. For traumatic brain injury, ablation of meningeal lymphatics aggravated neuroinflammation[34]. In the current study, we found that efflux of ISF macromolecules in the mPFC and the VTA of female mice were impaired by SCVS. Furthermore, results from lipidomic analyses as well as intracerebral injection of TAGs suggest that SCVS induced accumulation of long-chain TAGs in the mPFC and the VTA, which can impair mPFC astrocytes and VTA dopaminergic neurons, whereas enhancement of meningeal lymphatics facilitates lipid efflux from the brain parenchyma. These findings are consistent with previous studies reporting that lymphatic vessels play an important role in transportation of lipids and that TAGs can modulate activity of VTA dopaminergic neurons[47,50]. However, it remains to be explored further the long-term impact of SCVS-induced accumulation of TAGs and other lipids in the mPFC and the VTA, whether these changes in lipidomic profiles are sex-dependent, and how excess TAGs contribute to the abnormalities in brain emotional circuit and thus increased susceptibility to depression.

Of note, we found that among the depression-like behaviors, the novelty suppressed feeding was not significantly affected by manipulation of the meningeal lymphatics. In line with several previous studies[82–84], these findings suggest heterogenous mechanisms underlying different depressive phenotypes. In addition, although improvement of meningeal lymphatics ameliorated SCVS-induced body weight loss and anxiety-like behaviors in female mice, impairment of meningeal lymphatics did not significantly impair body weight or induce anxiety-like behaviors in stressed male mice. These findings

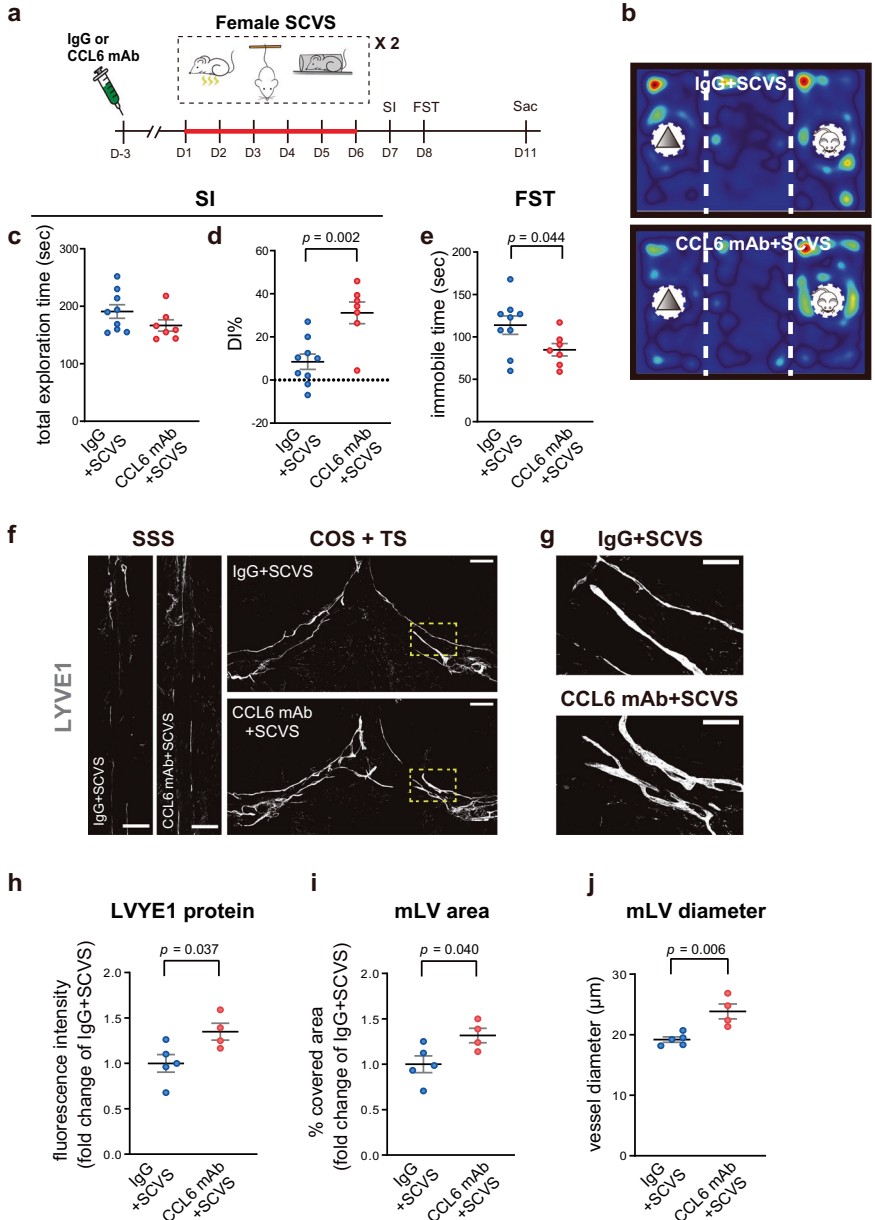

**Fig. 8 | Intracisternal delivery of CCL6-neutralizing antibody (CCL6 mAb) alleviates sub-chronic variable stress (SCVS)-induced depression-like behaviors and impairment of meningeal lymphatics (mLV) in female mice. a** The experimental timeline of the intracisternal CCL6 mAb or control immunoglobulin (IgG), SCVS paradigm, behavioral tests, and tissue collection (sac). **b** Representative heatmaps of animals' trace in the social interaction (SI) test. **c**, **d** Quantification of the total exploration time and differential index (DI) in the SI, comparing female mice intracisternally injected with CCL6 mAb or control IgG. Both groups of mice experienced SCVS. **e** Quantification of the immobile time in the forced swim test (FST) (**c**–**e**: *n* = 7–9 per group; results from two independent experiments). **f** Representative images of the LYVE1 staining (gray) in the SSS and COS + TS areas of dura mater. Scale bars: 500 μm. **g** Representative images of LYVE1-labeled mLV at higher magnification. Scale bars: 200 μm. **h**–**j** Quantification of the fluorescence intensity of the LYVE1 staining (**h**), the area covered by mLV (**i**), and the diameter of LYVE1-labeled mLV (**j**) in the SSS and COS + TS areas of dura mater (**h**–**j**: *n* = 4–5 per group). All data are presented as mean ± s.e.m. and analyzed by unpaired Student's *t* tests. Source data are provided as a Source data file.

suggest that impairment of meningeal lymphatics is required but not sufficient for stress-induced body weight changes and anxiety-like behaviors. Alternatively, the inconsistency may be explained by differential sensitivity of female and male mice to the manipulations of meningeal lymphatics. Moreover, despite significant reduction in the mPFC astrocytes and c-FOS expression in the dopaminergic neurons of the VTA were induced by impairment of meningeal lymphatics or blockage to drainage pathway to the dCLNs in male mice subjected to SCVS, the alteration in depression-like behaviors was selective to certain behavioral tests. We reasoned that there might be two factors underlying the phenotypic inconsistency between the detected

changes in the brain regions and the behavioral outcomes. First, more prolonged or larger magnitude of changes in the mPFC astrocytes and VTA dopaminergic neuronal activity may be required to generate robust depression-like behaviors in male mice[7,11]. Second, out results suggest that manipulation of meningeal lymphatics per se does not directly alter depression-like behaviors in mice, but rather changes their susceptibility to stress. Of note, although 6 days of SCVS can induce depression- and anxiety-like phenotypes in female mice, 21 days of chronic variable stress was required in order to generate robust behavioral changes in male mice[64]. Therefore, it is possible that administration of variable stress with prolonged period of time is

required for impairment of meningeal lymphatics to generate more depression-like phenotypes in male mice.

Our findings have therefore promoted a better understanding of the mechanisms underlying sex differences in depression. Despite of the strong sexual dimorphism in depression documented in clinical studies[3–6], the investigation of the underlying mechanisms is still at very early stage. Previous studies have revealed sex differences in several aspects of the brain parenchyma, including changes in the volumes, abnormalities in the electrophysiological properties of neurons and transcriptomic profiles of several brain regions including the mPFC and the VTA, as well as serotonin signaling and hypothalamic–pituitary–adrenal axis activity by stress[35,64,85–87]. Our study has identified that in addition to changes in the brain parenchyma, the sex different sensitivity of meningeal lymphatics to stress is an aggravating factor, and contributes to, at least in part, sex-dependent changes in the brain parenchyma and mood modulation. Much more work is still required to understand the complex relationship between sex different properties of the meningeal lymphatics with those in the brain parenchyma during the development of depression.

We would like to point out that the drainage function of the meningeal lymphatics can be assessed by clinically available imaging techniques, such as magnetic resonance imaging (MRI) and computed tomography (CT)[26,32,88,89]. It will therefore be worth examining in the future whether abnormalities in the meningeal lymphatics can serve as a novel biomarker for early diagnosis and prediction for the effectiveness of depression treatment in the clinics. Our study has also suggested that the improvement of meningeal lymphatics may be of therapeutic potential for depression treatment and other stress-related neuropsychiatry disorders.

# Methods

## Animals
Three-month-old female and male wild-type C57BL/6J mice obtained from the Institute of Experimental Animals of Sun Yat-sen University, Guangdong Medical Laboratory Animal Center or Zhuhai Bai Shi Tong Animal Center, and female CX3CR1-GFP transgenic mice (stock #005582; the Jackson Laboratory), were used for the experiments. The sex of animals used in each experiment was specified in the related texts of the result section and figure legends. The animals were housed in groups of 4–5 in an environmentally controlled (temperature: $23 \pm 2\,°C$, humidity: 50–60%) animal facility on a 12 h/12 h dark/light cycle, with access to regular food and water ad libitum. The animals were allowed to habituate to the animal facility for at least 1 week before the start of experiments. All animal studies were reviewed and approved by the Institutional Animal Care and Use Committee of Sun Yat-sen University (approval number: SYSU-IACUC-2019-000010).

## Sub-chronic variable stress (SCVS)
Animals were handled for 2–3 min per day for 3 days before the experiments. They were then subjected to three different stressors, alternating over a period of 6 days to avoid habituation: On day 1, the mice (5–10 together in the shock chamber, Med Associates, St. Albans, Vermont, USA) received one hundred 0.45 mA, 1 s foot-shocks delivered at random intervals for 1 h. On day 2, they were subjected to tail suspension stress for 1 h. On day 3, the mice were each placed inside a perforated restraint tube for 1 h within their home-cage. The same stressors were repeated for the next 3 days. The body weights were measured before the first day of stress and 1 d after the last day of stress to calculate the stress-induced changes in body weight.

## Behavioral tests
Depression-like and anxiety-like behaviors were assessed over 2 days after the completion of SCVS. Prior to testing, the animals were allowed to acclimate to the testing room for 1–2 h. The behavioral testing apparatus was cleaned with 75% ethanol between animals. The order of the behavioral tests was specified in the experimental timeline in the figure legends.

**Novelty suppressed feeding.** After food restriction for 24 h, mice were placed into one of the corners of a white plastic box ($40 \times 40 \times 40$ cm) with a single food pellet present in the center. The box was enclosed in a sound-insulated chamber, illuminated under red-light (80–100 Lux). The latency to begin food consumption in a 10-min test was measured by a researcher blinded to the experimental groups.

**Splash test.** The test was carried out in a normal mouse-cage under red-light (80–100 Lux) in a sound-insulated chamber. The dorsal coat of mice was sprayed 3 times with a 10% sucrose solution and video-taped for 5 min. The time spent grooming was recorded using a stopwatch by a researcher blinded to the experimental groups.

**Forced swim test.** Mice were placed into a 5 L glass beaker containing 3.5 L of water at 24–25 °C under white light (210–280 Lux), and videotaped for 6 min. The immobility time, defined as absence of any movement except that necessary for the mice to keep their heads above water, was scored over the last 4 min duration by a researcher blinded to the experimental groups.

**Open field.** The open-field arena was a white box ($40 \times 40 \times 40$ cm) illuminated with red-light (80–100 Lux), placed inside a sound-insulated chamber. The mice were placed onto a corner of the arena and allowed to explore the arena for 5 min. The total distance traveled, as well as the distanced traveled and time spent in the $20 \times 20$ cm center zone were tracked and analyzed by the DigBehv software (version 4.1.7.171129, Ji-Liang, Shanghai, China).

**Social interaction test.** Social interaction was measured in a $60 \times 40 \times 20$ cm box, divided into three chambers by two transparent dividing walls, under regular white light (210–280 Lux). For each dividing wall, there was an opening ($5 \times 5$ cm) located in the center which allowed free access of mouse to each chamber. Two wire-mesh enclosures were placed in the middle of the left and right chambers. A stranger mouse was habituated in the enclosure for 10 min one day before testing to minimize excessive stress. During the test day, the test mouse was firstly allowed to explore the box with empty enclosures for 10 min. Then the stranger mouse was placed in one of the enclosure, while a toy mouse was placed in the other enclosure. The test mouse was placed back to the box for 10 min free exploration. Both phases of the test were videotaped and analyzed by a researcher blinded to the experimental groups. Heatmaps of mouse' traces were generated by the EthoVision XT software (version 15.0.1416, Noldus, Netherlands). The discrimination index (DI%) was calculated as follows: (time exploring the stranger mouse − time exploring the toy mouse)/ (time exploring the stranger mouse + time exploring the toy mouse) × 100%.

**Sucrose preference test.** During this test, each mouse was individually housed in a cage with two bottles with drinking water. All mice had access to chow diet ad libitum. On day 1, the mouse was adapted to the environment and drank freely for 24 h. On day 2–4, a bottle of regular drinking water (M1 water) and a bottle of 0.5% sucrose solution (M1 sugar) were placed in the mouse cage. The position of two bottles were exchanged daily to minimize possible place-preference, and measurements of consumed water and sucrose solution were recorded daily. On day 5, the weight of the remaining water (M2 water) and 0.5% sucrose solution (M2 sugar) were recorded. At the end of the test, all animals were returned to group housing. Sucrose

preference was calculated as follows: [M1 sugar − M2 sugar]/ [(M1 sugar − M2 sugar) + (M1 water − M2 water)] × 100%.

## Corticosterone administration treatment

For chronic corticosterone exposure to induce depression-like behavior, male mice received intraperitoneal injection of corticosterone (40 mg/kg body weight, dissolved in 10% (2-Hydroxypropyl)-β-cyclodextrin in saline, Sigma, cat. No. 27840) or vehicle once daily for 33 days[90]. Vehicle-treated control animals received injections of 10% (2-Hydroxypropyl)-β-cyclodextrin in saline in parallel. Injections were given between 9:00 and 11:00 AM. On day 29 and 30 of injection, mice underwent the behavioral tests beginning 1 h after receiving corticosterone or vehicle injection, and were sacrificed for tissue collection on day 33.

## Measurement of serum corticosterone level

Orbital sinus blood was collected under anesthesia at 8:00 AM and 8:00 PM on the day before the beginning of SCVS (before SCVS), and the day after the completion of SCVS (after SCVS). For each collection, ~100 μL whole blood was collected into 0.5 mL tube, sat for 1 h at room temperature for clotting, and centrifuged for 15 min at $956 \times g$. The supernatants were collected as serum and stored at −80 °C until analysis. Serum corticosterone concentrations were measured with an ELISA kit (Abcam, Ab108821), following the manufacturer's instructions.

## Intra-cisterna magna injection

Seventy-two hours after the last behavioral test (unless specified in the "Results" section), the animals were injected intracisternally with a fluorescent tracer and sacrificed 1 h later to evaluate the drainage function of meningeal lymphatics. We chose the 72 h post-testing (5 days after the last stress episode) as the time point for tissue collection for two reasons: first, to avoid potential complication of acute stress from behavioral testing on animals; and second, to examine whether SCVS could induce relatively long-lasting changes in the meningeal lymphatics.

Mice were anesthetized by sodium pentobarbital (100 mg/kg body) and placed on a heating pad to maintain body temperature. Their eyes were lubricated with Vaseline mixed with sterile saline. The head of mouse was secured on a stereotaxic frame. The hair over the back neck was shaved and the underneath skin were disinfected with iodine and 70% ethanol. An incision was made at the midline, and muscle layers were retracted to expose the cisterna magna. The injection was carried out using a Hamilton syringe mounted on an infusion pump (Model R452, RWD Life Science, China) and coupled to a 30-gauge needle by a PE tube. 5 μL 70 kDa Dextran-Texas Red or Ovalbumin-Alexa Fluor 647 (Thermo Scientific, Waltham, MA, USA; 0.5% in artificial cerebrospinal fluid) were injected at a rate of 1 μL/min. The needle was left in place for an additional 10 min before withdrawal. The skin was sutured. All mice were kept anesthetized on a heating pad before being euthanized at time specified in the experimental timeline in the figure legends after the end of tracer injection. Intracisternal AAV infusion was carried out in a similar way. 3 μL AAV1-CMV-VEGFC, AAV1-CMV-eGFP, AAV9-CAG-VEGFR3$_{d1-4}$-IgG Fc, AAV9-CAG-VEGFR3$_{d4-7}$-IgG Fc, or AAV9-CAG-eGFP ($1 \times 10^{12}$ genome copies per mL, OBiO, Shanghai, China) was injected[30,39]. Mice were allowed to recovered for 28 days before conducting SCVS experiments. For intracisternal CCL6-neutralizing antibody infusion, rat anti-CCL6 IgG (MAB487, lot# UPR0221021, R&D System, USA) and rat isotype control IgG2b (MAB0061, lot# HBI1021101, R&D System, USA) were used[65]. 5 μL antibody (1 μg/μL) was injected at a rate of 1 μL/min and the needle was left in place for an additional 10 min before withdrawal. Mice were allowed to recover for 3 days before conducting SCVS experiment.

## AAV viral vector construction

For AAV viral vectors used in this study, the AAVs expressing eGFP (AAV-CMV-eGFP-3FLAG, AAV-CAG-eGFP), VEGFC (AAV-CMV-VEGFC-P2A-eGFP-3FLAG), and two VEGFR3 mutants (AAV-CAG-VEGFR3$_{d1-4}$-IgG Fc and AAV-CAG-VEGFR3d$_{4-7}$-IgG Fc) were custom-made by OBiO (Shanghai, China). Briefly, PCR-amplified fragments of mouse *Vegfc* coding sequence (NM_009506.2) were cloned into the pAAV-CMV-eGFP-3FLAG backbone vector (CMV: cytomegalovirus promoter; eGFP: eGFP, enhanced green fluorescent protein; 3FLAG: 3 copies of FLAG epitope tag) with self-cleaving 2A peptide (P2A) in between sequences encoding VEGFC and eGFP (AAV-CMV-VEGFC-P2A-eGFP-3FLAG)[30,67]. PCR-amplified fragments of mouse VEGFR3-coding sequence (NM_008029.3) encoding amino acid sequence 45–415 of mouse VEGFR3 domain 1–4 (VEGFR3$_{d1-4}$) and amino acid sequence 331–764 of VEGFR3 domain 4–7 (VEGFR3$_{d4-7}$) were cloned into the pAAV-CAG-IgG Fc-HA backbone vector and fused in frame with the mouse IgG Fc domain (CAG: the CMV early enhancer/chicken beta-actin promoter; IgG: mouse IgG Fc domain; HA: the hemagglutinin epitope tag) for constructing the pAAV-CAG-VEGFR3$_{d1-4}$-IgG Fc and pAAV-CAG-VEGFR3$_{d4-7}$-IgG Fc plasmids, as reported previously[39,91]. The plasmids of AAV-CMV-eGFP-3FLAG and AAV-CMV-VEGFC-P2A-eGFP-3FLAG were packaged into AAV serotype 1 (AAV1). The plasmids of AAV-CAG-VEGFR3$_{d1-4}$-IgG Fc, AAV-CAG-VEGFR3$_{d4-7}$-IgG Fc, and AAV-CAG-eGFP were packed into AAV serotype 9 (AAV9).

## Analysis of distribution of intracisternally-injected fluorescence tracer in the brain parenchyma

One hour after the intracisternal injection of fluorescence tracer, mice were anesthetized and transcardially perfused with PBS for 1 min and 4% paraformaldehyde for 5 min at a rate of 10 mL/min. The brains were harvested and postfixed in 4% paraformaldehyde at 4 °C overnight. 100 μm coronal brain sections were collected by a cryostat, stained with DAPI (5 μg/mL, Cell Signaling Technology) in PBS for 15 min at room temperature, and mounted in anti-fade reagents on the glass slides. The infiltration of the tracer was imaged by Nikon Eclipse Ni-U epi-fluorescent microscope (Nikon, Japan). The region of interest (ROI) of 8 brain slices (as specified in the related figure legends) per animal was defined based on the DAPI channel, according to the coronal mouse brain atlas provided by the Allen Brain Institute[92], and tracer coverage between slices were analyzed by ImageJ (version 1.52n, NIH, USA) and averaged as a biological replicate for one animal. The analyzed ROIs include primary motor cortex (M1), medial prefrontal cortex (mPFC), striatum (STR), thalamus (THAL), hypothalamus (HYP), hippocampus (HPC), ventral tegmental area (VTA), and periaqueductal gray (PAG).

## Evans blue injection

After the female mice underwent 6 days of SCVS, mice were intracisternally injected with 1% Evans blue dye (Sigma) in the artificial cerebrospinal fluid and sacrificed 1 h later for brain collection. The whole brain (including cerebellum) was homogenized by a Glas-Col homogenizer on ice, let stand on ice for 30 min (protected from light), and centrifuged at $21,130 \times g$ for 30 min in 4 °C. Evans blue dye in the supernatant was then measured by a multi-functional plate reader (Victor Nivo 5s, PerkinElmer, USA) at 620 nm and converted into milligram dye per milligram tissue[93].

## Lymphatic vessel ligation

The mouse was anesthetized. The hair over the front neck was shaved and the underneath skin was disinfected with iodine and 70% ethanol. Their eyes were lubricated with Vaseline mixed with sterile saline. An incision was made at the midline, and muscle layers were retracted to expose the dCLNs. Under a stereo-microscope (RWD Life Science, China), the collecting lymphatic vessels anterior to the dCLNs were ligated by 8-0 nylon sutures. Sham-operated mice received the incision

and had the muscle layer retracted, but the lymphatic vessels were not ligated. After the surgery, the skin was sutured and mice were placed on a heating pad until awake. Mice were allowed to recover for 7 days before SCVS experiments.

## Immunofluorescence staining

Mice were transcardially perfused with PBS for 2 min and 4% paraformaldehyde for 5 min at a rate of 10 mL/min. The whole dorsal meninges attached to the skullcap were harvested[25,31], and postfixed in 2% paraformaldehyde at 4 °C overnight. The dura mater was then carefully peeled off from the skull. The brains and dCLNs were harvested[25,31], and postfixed in 4% paraformaldehyde at 4 °C overnight, cryoprotection in 30% sucrose in PBS for 2–3 days at 4 °C. 30–40 μm coronal brain sections were collected by a cryostat for free-floating staining. Two coronal sections containing the mPFC at approximately 1.77 and 1.97 mm anterior to the Bregma, and two sections containing the VTA at approximately 2.91 and 3.15 mm posterior to the Bregma were sampled for each immunostaining. The averaged values for the same brain region of each animal were taken as one data point. The brain regions of interest were identified based on the 4th edition of "the Mouse Brain in Stereotaxic Coordinates" by Paxinos and Franklin[94]. The sections were incubated with a blocking buffer (5% goat serum, 1% BSA, 0.4% Triton X-100 in PBS) for 2 h at room temperature, and then with primary antibodies (diluted in the blocking buffer) for GFAP (Abcam, Cambridge, UK; cat# ab4674, lot# GR3349945-1; 1:5000), S100β (Abcam, Cambridge, UK; cat# ab52642, lot# GR3215095-15, clone# EP1576Y; 1:500), TH (R&D Systems, Emeryville, CA, USA; cat# MAB7566, lot# CGUO0120071, clone# 779427; 1:8000) and c-FOS (Cell Signaling Technology, Danvers, MA, USA; cat# 2250, lot# 12, clone# 9F6; 1:500) for 48 h at 4 °C. Afterward, the sections were washed for 3 times of 10 min each in PBS with 0.4% Triton X-100, followed by incubation with secondary antibodies diluted in the blocking buffer for 2 h at room temperature. The sections were washed again for 3 times and mounted onto glass-slides. The schematic diagram of dura mater was referenced from previous reports[25]. For whole-mount meninges staining, the meninges were blocked with PBS containing 5% goat serum, 1% BSA, 0.1% Triton X-100, and 0.05% Tween-20 for 2 h at room temperature, and incubated with primary antibody for LYVE1 (Abcam, Cambridge, UK; cat# ab14917, lot# GR3392340-6; 1:500) diluted in PBS containing 1% BSA, 0.5% Triton X-100 for overnight at 4 °C. Washes and secondary antibody staining were carried out similarly to the staining procedures of brain sections. 30-μm sections of dCLNs were collected, mounted on glass-slides. Images were taken by Nikon Eclipse Ni-U epi-fluorescent microscope (Nikon, Japan) or Zeiss LSM880 confocal microscope (Zeiss, Germany), and analyzed by ImageJ (version 1.52n, NIH, US). For assessment of the diameter of meningeal lymphatic vessels, 80 measurements (40 along the TS and 40 along the SSS) per meninx were taken by an experimenter blinded to the experimental groups, and the averaged value was taken as one data point[34]. For assessment of i.c.m.-injected tracer diffusion in the brain parenchyma, 8 representative brain sections were imaged by the Nikon Eclipse Ni-U epi-fluorescent microscope (Nikon, Japan) equipped with a motorized stage and the images of each brain section were automatically stitched. For assessment of tracer in the dCLNs, 8–10 representative sections from each of left and right dCLNs were imaged and the mean area fractions were calculated for each animal.

## Phagocytosis of meningeal tracer

For the assessment of the proportion of OVA-647 tracer phagocytosed by the macrophages on the meninges, female CX3CR1-GFP transgenic mice at 3 months of age were used as previously described[29]. Female CX3CR1-GFP mice went through SCVS for 6 days, followed by intra-cisternal OVA-647 injection and 1 h later, mice were sacrificed for meninges collection and staining following the same procedures as described in the section above. The proportion of OVA-647 phagocytosed by the GFP-labeled macrophages in the tracer-enriched TS region of the meninges was imaged by Zeiss LSM800 confocal microscope with ×10 objective lens. The ratio of OVA-647 covered area in the total imaged areas and in GFP-labeled macrophage area was analyzed by ImageJ (version 1.52n, NIH, USA).

## Tissue dissection and metabolite extraction

Mice were sacrificed by rapid cervical dislocation. The brains were sliced into 1 mm sections on a brain matrix (RWD Life Science, Shenzhen, China) in ice-cold dissection buffer. The mPFC and VTA were dissected out with a 15G puncher. The meninges were scrapped from the skullcap in ice-cold dissection buffer. The sample was snapped frozen on dry ice. Four meninges or indicated brain regions were pooled as one biological sample, and 3 biological samples per group were used for metabolite extraction and LC-MS analysis, which were carried out by BIOTREE (Shanghai, China). 25 mg of each sample was added sequentially with 200 μL water and 480 μL extract solution (methyl-tert-butyl ether/methanol = 5:1, vol/vol). After vortexing for 30 s, the samples were homogenized by sonication for 10 min in ice-water bath. The homogenization and sonication cycles were repeated for 3 times, followed by incubation at −40 °C for 1 h and centrifuged at $845 \times g$ for 15 min at 4 °C. 300 μL of supernatant was transferred to a fresh tube and dried in a vacuum concentrator at 37 °C. The dried samples were then re-constituted in 100 μL solution of dichloromethane/methanol mixture (1:1, vol/vol) by vortexing for 30 s and sonication for 10 min in ice-water bath, and centrifuged at $15,871 \times g$ for 15 min at 4 °C. 75 μL of supernatant was transferred to a fresh glass vial for LC/MS analysis. The quality control (QC) sample was prepared by mixing an equal amount of 20 μL supernatants from all samples.

## LC-MS/MS lipidomic analysis

LC-MS/MS analyses were performed using an UHPLC system (1290, Agilent Technologies), equipped with a Kinetex C18 column (2.1 × 100 mm, 1.7 μm, Phenomen). The mobile phase A consisted of 40% water, 60% acetonitrile, and 10 mmol/L ammonium formate. The mobile phase B consisted of 10% acetonitrile and 90% isopropanol, which was added with 50 mL 10 mmol/L ammonium formate for every 1000 mL mixed solvent. The analysis was carried out with elution gradient as follows: 0–1.0 min, 40% B; 1.0–12.0 min, 40–100% B; 12.0–13.5 min, 100% B; 13.5–13.7 min, 100–40% B; 13.7–18.0 min, 40% B. The column temperature was 55 °C. The auto-sampler temperature was 4 °C, and the injection volume was 2 μL (pos) or 2 μL (neg), respectively. The raw data files were converted to files in mzXML format using the 'msconvert' program from ProteoWizard. The CentWave algorithm in XCMS was used for peak detection, extraction, alignment, and integration. The minfrac for annotation was set at 0.5, and the cutoff for annotation was set at 0.3. Lipid identification was achieved through a spectral match using LipidBlast library, which was developed using R and based on XCMS (version 3.2). The Partial least squares discrimination analysis (PLS-DA) was used to determine lipid differences between the three groups. For paired comparison, the score plots of Orthogonal Partial Least Squares-Discriminant Analysis (OPLS-DA) model was generated on the positive- and negative-mode MS data obtained from the XCMS program, which had been centered and scaled to unit variance scaling. Both PLS-DA and OPLS-DA were performed using the SIMCA (V16.0.2, Sartorius Stedim Data Analytics AB, Umea, Sweden). The significantly altered metabolites were determined by variable importance in projection (VIP) scores from pairwise OPLS-DA analysis and comparisons using the Student's t-test. Differentially changed lipid were defined as those with Student t-test p value <0.05 and the variable importance in the projection (VIP) >1, and visualized as heatmaps using the pheatmap (version 1.0.12) package in R (version 4.1.0).

## Intracerebral triacylglycerol (TAG) and tracer injection

Synthetic TAG powder (Macklin, Cat. no. G810370, China) was dissolved in chloroform (200 μL chloroform per 100 mg TAG), dried by nitrogen gas, and re-constituted in BSA solution (10 mg/mL in PBS) with sonication, to make 100 mg/mL TAG stock solution. Mice were anesthetized and head-fixed on a stereotaxic frame. mPFC (A/P: +1.8 mm, M/L: ±0.3 mm, D/V: −2.45 mm) and VTA (A/P: −3.2 mm, M/L: ± 0.5 mm, D/V: −4.45 mm) were injected with 1 mg/mL TAG (diluted in sterile PBS) or vehicle (0.1 mg/mL BSA in sterile PBS) at a volume of 200 nL and injection rate of 40 nL/min, and the needle was left in place for an additional 5 min before withdrawal. Mice were sacrificed at 24 h after injection and their brains were collected for immunofluorescence staining as described in above. The same procedure was used for intracerebral tracer injection. 200 nL mixture of 70kD Dextran-Texas Red and 40kD Dextran-Alexa Fluor 488 (Thermo Scientific, Waltham, MA, USA; 0.01% in sterile PBS) were injected into mPFC and VTA. Mice were sacrificed 2.5 h after tracer injection.

## RNA extraction and quantitative real-time polymerase chain reaction (qPCR) analysis

Mice were euthanized by rapid cervical dislocation. The meninges were quickly scrapped from the skullcap in ice-cold dissection buffer, and snap-frozen on dry ice. Total RNAs were extracted using Trizol (Thermo Scientific, Waltham, MA, USA) and the cDNAs were synthesized using the NovoScript Plus All-in-on Strand cDNA Synthesis Supermix (Novoprotein, Suzhou, China) following the manufacturer's instructions. For qPCR analysis, the following primers were used: *Lyve1* primers (forward: 5′-CAGCACACTAGCCTGGTGTTA-3′, reverse: 5′-CGCCCATGATTCTGCATGTAGA-3′), *Gapdh* primers (forward: 5′-GAA-CATCATCCCTGCATCCA-3′, reverse: 5′- CCAGTGAGCTTCCCGTTCA-3′). 5 ng cDNA were amplified with NovoStart SYBR qPCR SuperMix Plus (Novoprotein, Suzhou, China) using the following condition: 95 °C for 3 min, followed by 40 cycles of 95 °C for 15 s, 60 °C for 30 s, and 72 °C for 20 s. Triplicates of each sample were analyzed, and the average cycle threshold (Ct) value was used for assessing the relative expression of target mRNAs using the ΔΔCt method. The relative amounts of *Lyve1*, mRNAs were normalized to the mRNA level of *Gapdh*.

## Sorting of meningeal LECs

Meningeal LECs were collected following previous reported protocol with modifications[43]. Mice transcardially perfused with ice-cold PBS for 1 min under anesthesia. Individual meninx was quickly dissected from the skullcap in DMEM with 2% FBS (Thermo Scientific, Waltham, MA, USA), and digested in preheated DMEM with 2% FBS, 1 mg/mL collagenase VIII, 0.5 mg/mL DNase I (Sigma-Aldrich, St. Louis, MO), 45 μM actinomycin D (Selleck, Houston, TX, USA) for 12 min at 37 °C. At the end of the digestion, 1 mL DMEM with 10% FBS were added to the solution to terminate digestion. Individual samples consisted of cell suspensions pooled from 3 to 4 meninges were filtered through a 70 μm nylon-mesh filter and washed with ice-cold fluorescence-activated cell sorting (FACS) buffer (PBS free of calcium and magnesium, 1 mM EDTA and 1% BSA, pH 7.4). The cells were pelleted by centrifugation at 400 × *g*, 4 °C for 5 min, resuspended in 400 μL FACS buffer with anti-CD45–BB515 (BD Biosciences, San Jose, California, USA; cat# 564590, lot# 0216978, clone 30-F11, 1:200), anti-CD31-Alexa Fluor 647 (BD Biosciences, San Jose, California, USA; cat# 563608, lot# 9199302, clone 390, 1:200), anti-Podoplanin-PE (eBioscience, San Diego, California, USA; cat# 12-5381-82, lot# 2120142, clone 8.1.1, 1:200) and DAPI (Sigma, 0.2 μg/mL), and incubated for 15 min at 4 °C. Cells were then washed and resuspended in ice-cold FACS buffer. Cells were sorted by MoFlo Astrios EQs (Beckman Coulter, Indianapolis, IN, USA) and analyzed by the FlowJo software (version 10, Tree Star, Ashland, OR, USA). Singlets were gated using the height, area and the pulse width of the forward-scatter cells, and cells negative for DAPI were selected as viable cells. The LECs were gated as CD45⁻CD31⁺podoplanin⁺ and sorted into a PCR tube containing 1 μL lysis buffer (Vazyme, Nanjing, China).

## Library preparation and RNA sequencing

The library preparation and RNA sequencing were carried out by GENEWIZ (South Plainfield, NJ, USA). The Discover-sc WTA Kit (version 6.2, Vazyme, Nanjing, China) was used to amplify the RNAs and prepare the cDNA library. Then the libraries were loaded onto an Illumina HiSeq instrument (Illumina, San Diego, CA, USA) for paired-end 150 bp sequencing. The raw sequencing reads were filtered for removing adapter sequencing, primers, polyA tail sequences, and reads of low-quality bases. The quality of the reads was evaluated using FastQC. Then the processed sequences were aligned to the mm9 mouse reference genome (UCSC) using Hisat2 (version 2.0.1). Low expression genes with count number no more than 5 in over 70% of the samples were excluded from analysis. DESeq2 (version 1.32.0) were used to normalize the raw counts and perform differential expression analysis. Differentially expressed genes (DEGs) were defined as having an adjusted *p* value <0.1 by the Benjanmini−Hochberg false-discovery rate procedure. The heatmap and volcano plot of the DEGs were generated using the pheatmap package (version 1.0.12) in R (version 4.1.0). Functional enrichments were performed with the gene-set enrichment analysis (GSEA) to identify KEGG pathways (c2.cp.kegg.v7.4.symbols.gmt) and gene ontology (GO, c5.go.bp.v7.4.symbols.gmt, c5.go.mf.v7.4.symbols.gmt and c5.go.cc.v7.4.symbols.gmt) that were differentially enriched in the stressed group and the control counterparts. Gene set size filters were set at minimum of 5 and maximum of 1000. False-discovery rates (FDR) for the enrichment score of the gene set were calculated based on 1000 gene set permutations. The top 10 gene sets enriched in each group were plotted with ggplot2 (version 3.3.5) in R (version 4.1.0). Gene expression datasets of different brain regions from woman MDD patients (Supplementary Table 3 of Labonte et al.[64]) and gene expression datasets of sorted meningeal LECs from aging mice (GEO accession number: GSE104181) were used for comparison with the DEGs identified in the current study. The VennDiagram (version 1.6.20) was used to identify common DEGs. Heatmaps were generated by the pheatmap package (version 1.0.12) of R (version 4.1.0) to show relative expression of the common DEGs in each biological replicate.

## Statistical analysis

Data were analyzed by Prism 7.0 (GraphPad Software, San Diego, CA) and presented as mean ± SEM. We conducted the Shapiro−Wilk test to evaluate the normality of the datasets. For datasets which were normally distributed, outliers were identified as being >2 standard deviations from the mean and excluded from statistical analysis as previously described[95]. For two-group comparisons, two-tailed unpaired Student's *t*-test was used for normally-distributed datasets, and the Mann−Whitney test was used for non-normally distributed datasets. For multiple-group comparisons, one-way or two-way ANOVA with Tukey's post hoc tests was used. $p < 0.05$ was considered as statistically significant.

## Reporting summary

Further information on research design is available in the Nature Research Reporting Summary linked to this article.

## Data availability

The RNA sequencing data generated for this study can be found in the GEO repository under accession number GSE201368. Source data are provided as a Source data file. Further information of this study is available upon reasonable request from the corresponding authors Xiaojing Ye (yexiaoj8@mail.sysu.edu.cn), Wei-Jye Lin (linwj26@mail.sysu.edu.cn), and Yan Zhang (yan.zhang@csu.edu.cn). Source data are provided with this paper.

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

## Acknowledgements

We thank Professor Ming-Hu Han, Dr. Hongxing Zhang, and Dr. Song Zhang at Icahn School of Medicine at Mount Sinai for technical advices on animal behavioral testing. We thank Professor Jonathan Kipnis, Dr. Justin Rustenhoven, and Dr. Sandro Da Mesquita at Washington University in St. Louis for technical suggestions on intracisternal injection and meningeal LEC sorting. We thank the animal facility and the core facility of the Zhongshan School of Medicine and the Sixth Affiliated Hospital, Sun Yat-sen University. We thank all members of the Ye lab and the Lin lab for discussion and technical assistance during the execution of this project. This work is supported by grants from the National Key Research & Development Program of China (No. 2021ZD0202000 to Y.Z. and X.Y.), National Natural Science Foundation of China (No. 81873797 to X.Y., No. 81972967 to W.J.L.), Science and Technology Program of Guangzhou (No. 202007030001 to X.Y. and W.J.L.), Guangdong Science and Technology Department (No. 2020B1212060018 and 2020B1212030004 to W.J.L.), Natural Science Foundation of Guangdong Province (No. 2019A1515011483 to X.Y., No. 2019A1515011754 to W.J.L.), the Fundamental Research Funds for the Central Universities (No. 19ykzd40 to X.Y.), and Guangdong Project (No. 2019QN01Y202 to X.Y.).

## Author contributions

X.Y. and W.J.L. designed the research project; W.D., M.Y., P.X., C.X., S.H., Z.Z., X.C., W.L., J.J., J.Z., W.B., and Y.Z. carried out the experiments; Y.Z., H.Z., Y.L., W.W. and P.W. provided reagents/materials and help analyzed the data; X.Y., W.J.L., W.D., M.Y., P.X., and Y.Z. wrote the manuscript. All authors contributed to the article and approved the submitted version.

## Competing interests

The authors declare no competing interests.
