## [Peer Review File · Nature Communications]

A functional role of meningeal lymphatics in sex difference of stress susceptibility in miceREVIEWER COMMENTS

Reviewer #1 (Remarks to the Author):

In this study, Dai et al explored whether meningeal lymphatic dysfunction contributes to stress induced depression. Using a sub-chronic variable stress (SCVS) paradigm to induce depressive-like behaviors, they show here that SCVS leads to impaired meningeal lymphatic drainage and altered lymphatic vasculature morphology in female but not male mice and that this correlates with the female-biased induction of depression-related behaviors in this SCVS model. To interrogate a mechanistic role for defective meningeal lymphatic drainage in the depressive phenotypes that develop in SCVS treated female mice, they showed that VEGFC-induced boosting of meningeal lymphatic drainage rescues the depressive phenotypes in female SCVS mice. Moreover, they further demonstrate that pre-existing meningeal lymphatic defects that are brought on with AAV-VEGFR3d1-4 pretreatment in male mice promotes the development of depression-associated behaviors following SCVS treatment. Interestingly, they also report that SCVS conditions lead to diminished activity of dopaminergic neurons in the ventral tegmental area and that this could be ameliorated with VEGFC-mediated boosting of meningeal lymphatic function. Overall, I am quite enthusiastic about this work and found their findings to be very exciting and of great potential significance. Despite these strengths, a few issues and technical shortcomings were noted, which I believe need to be addressed in order to solidify their central findings and conclusions.

Major comments:

1. A few important controls are missing throughout the manuscript. For instance, it will be important to show that meningeal lymphatic drainage is not altered in SCVS treated male mice in comparison to non-stressed male controls (similar to the studies done in SCVS females in Figure 1n). It will also be important to evaluate if VEGFC treatment alone in non-stressed female mice impacts any of the behaviors presented in Figure 2l-r. Likewise, it is also important to provide non-stressed male controls for the behavior studies exploring the effects of lymphatic ablation (AAV-VEGFR3d1-4 treatment) in Figure 4k-r.
2. It is unclear based on the data provided in Figure 3 whether CSF influx and ISF diffusion (glymphatic function) are altered in their SCVS model. In Figure 3a, they appear to show glymphatic drainage of OVA tracer in the brain parenchyma. However, they do not go on to investigate whether this is affected by SCVS or if this can be modulated by VEGFC treatment. These specific studies should be conducted to address this issue and to also support their suggestive astrocyte end-feet data presented in Figure 3. Importantly, they should also provide quantification of the OVA tracer glymphatic drainage in addition to the representative images presented in Figure 3a.

Minor comments:

1. In line 93, it should be "interleukin-6" and not "interleukine-6".
2. In line 94, I think it should be "reverse fashion" and not "reversed fashion".

Reviewer #2 (Remarks to the Author):

In this manuscript, the authors are describing a role of meningeal lymphatic vasculature in a model of sub-chronic variable stress that only leads to depressive-like behavior in females. Experimental data using this sex biased stress model in combination with gain and loss of function manipulations of meningeal lymphatic vasculature suggest that modulation of meningeal lymphatic drainage can be considered a therapeutic target in depression. However, all the data is correlative and does not imply a direct involvement of the meningeal lymphatic system, or a mechanism through which reduced (or enhanced) meningeal lymphatic drainage is underlying the described phenomena in the context of stress-induced depressive-like behavior. There are several concerns that need to be addressed regarding some of the described concepts, the current provided data, and data interpretation. There are also additional experiments and data that need to be provided to allow the authors to fully substantiate their claims/conclusions.

Specific comments/concerns/experimental requests:

- The authors refer (at least twice) that the meningeal lymphatics drain “excess” fluid, macromolecules and immune cells. This is a wrong description of the process. Lymphatic vessels in the meninges constantly, and rather unselectively, drain CSF molecules that reach the dural stroma. Molecular content from the CSF is constantly drained into the cervical lymph nodes – not only in situations where molecules are “in excess”. The process of immune cell drainage is quite different. This is not a passive, but active process, that depends on chemokine gradients alongside other mechanisms. The authors need to be thorough when describing these processes.

- Measurement of Lyve1 mRNA in the dura matter (Fig. 1a and b) cannot be used as a surrogate of lymphatic vasculature function. There is a very well described population of resident dural macrophages that express high levels of Lyve1 mRNA and LYVE-1 protein.

- What are the differences between basal adult male and female meningeal lymphatic vessel morphology and drainage capacity to the deep CLNs? This analysis should be performed.

- Why are females, but not males, affected using this stress protocol? Authors have to provide more data about this. One suggestion is to measure the levels of corticosterone during day and night in females and males from control and SCVS groups. This will provide additional clues on why meningeal lymphatics might be affected in females but not males.

- Are different levels of corticosterone in stressed females and males responsible for the different phenotypes in terms of meningeal lymphatic function. What is the mechanism underlying the loss of meningeal lymphatic coverage in stressed females?

- The results in Fig. 1g and I suggest that less CSF is reaching the vicinity of the meningeal lymphatics, which means that the deficits in the amount of tracer that is reaching the dCLN is not due to impaired drainage by meningeal lymphatics, but by a potential impairment in the pathway that connects the brain perivascular/subarachnoid spaces (that become filled with the injected tracer) and the dural stroma. Similar results are also observed in the data in Fig. 4c and h, where stressed female mice treated with VEGFR3-(1-4) also show less fluorescent tracer (hence less CSF) reaching the dural compartment and the draining lymphatic vessels. The opposite is observed in Fig. 2c and h, where VEGF-C leads to more fluorescent tracer reaching the dural tissue. Most of the fluorescent tracer that reaches the dura in postmortem tissue is present inside dural macrophages (Rustenhoven et al, Cell 2021). Another possible explanation is that the dural macrophages in the stressed female present deficits in phagocytic capacity. In sum, these observations put into question the loss of drainage capacity by meningeal lymphatics in the context of SCVS. The authors need to complement the measurements of fluorescent tracer in the dura with measurements of fluorescent tracer left in the CSF (collected from the cisterna magna or the lateral ventricle) at the time of euthanasia (1 h post icm injection). It is also very important to perform experiments to assess the activation status of dural myeloid cells (e.g., macrophages), by flow cytometry or other techniques, to discard (or consider) the involvement of macrophages (and their secreted cytokines) in the described phenomena.

- The control for the VEGFR3-(1-4) viral vector should be injection of an AAV expressing the domains 4-7 of the receptor (coupled to a Ig or Fc domain), and not eGFP. It is not clear where the authors manufactured or purchased the AAV constructs. Are they commercially available? If this is the case, they need to provide the references. If this is not the case, more details must be provided on how the viral constructs were generated. This is mandatory to allow reproducibility of the experimental results.

- What is the effect of loss of meningeal lymphatic function (upon VEGFR3-(1-4)) or gain of meningeal lymphatic function (upon VEGF-C) on the performance of female and male (without the influence of stress) mice in the ST, FST, NSF? It is possible that defective meningeal lymphatic drainage per se is enough to cause depressive-like behavior in males and/or females. To my knowledge, this has never been explored. These controls are missing in all the data presented in Figs 2 and 4. For the sake of the overall message of the study, it is imperative that these experiments are performed, and this data is included in the manuscript.

- Treatment of stressed female with VEGF-C seems to result in better meningeal lymphatic drainage, alongside with more S100b+ and GFAP+ cells in the mPFC and c-FOS+ cells in the VTA. These observations are correlated but have not been functionally linked by the data provided by the authors. VEGF-C has the capacity to signal on other cells in the brain that express VEGFR2 or VEGFR3, hence the observed effects might well be lymphatic-independent. The authors need to provide experimental evidence that points to a potential mechanism linking increased lymphatic drainage, due to VEGF-C, to the described region-specific alterations in glial and neuronal responses.

- Throughout the manuscript there is no reference on whether the results have been replicated in independent experiments.

Reviewer #3 (Remarks to the Author):

Women are more likely to experience major depressive disorder (MDD) and severe symptoms, however, underlying biological mechanisms that could explain these sex differences are still unclear. In recent years, neuroimmune mechanisms of depression have received increasing attention. Nevertheless, the potential role of the meningeal lymphatic system, which is essential to drain excess macromolecules from the brain parenchyma, has yet to be explored. Here, the authors took advantage of the subchronic variable stress (SCVS) mouse model of depression to investigate efficiency of brain drainage function following stress exposure in a sex-specific manner. Indeed, the SCVS paradigm induces depression-like behaviors in female but not male mice allowing characterization of sex differences. To confirm that stress-induced alterations in meningeal lymphatics structure and function are involved in the establishment of anxiety- and depression-like behaviors, experiments were performed using a viral-mediated strategy to overexpress vascular endothelial growth factor C (vegfc, gain-of-function) or vegf receptor 3 mutants (loss-of-function). These manipulations affect activation of dopaminergic neurons in the ventral tegmental area (VTA) and astrocytes in the medial prefrontal cortex (mPFC). Finally, RNA-seq analysis were performed on sorted meningeal lymphatic endothelial cells revealing altered transcriptomic profiles after SCVS exposure in female mice.

Overall, the question addressed in this study is intriguing, findings are novel, and the manuscript is clear and well written. Figures are of high quality and very informative. Nevertheless, I do have a number of concerns that need to be addressed, as listed below, as I do believe that the results presented do not accurately and convincingly support the general conclusion that meningeal lymphatics are involved in stress susceptibility or depression in a sex-specific manner.

Major:

- All experiments were performed in mice only, but the title and last sentence of the abstract are referring to human depression. This should be modified to better reflect the findings.

- Chronic stress is the main environmental factor to develop human depression. Although widely used to study sex differences in rodent with regard to stress responses, the SCVS paradigm is not considered a model of chronic stress. As such, the authors should be careful when extrapolating SCVS results to human depression. Moreover, findings described may not reflect the state of meningeal lymphatics after chronic stress exposure or at later time points after SCVS. These limitations should be highlighted in the Discussion.

- Characterization of LYVE1 protein level, diameters of LYVE1-labelled meningeal lymphatic vessels, coverage, and level of injected tracers in the SS, TS and COS should be performed in unstressed and stressed males to confirm sex-specific changes in brain drainage function. Lack of change in mRNA expression do not always correlate with unaltered protein structure or function. Current data do not support that SCVS induced structural and functional impairment of the meningeal lymphatics in female, but not male mice as stated in the last sentence of this section.

- In Figure 2, eGFP+SCVS and VEGFC+SCVS are compared. Were unstressed mice injected with the eGFP and VEGFC viruses as well? Assessment of baseline biological and behavioral effects of VEGFC overexpression would strengthen the manuscript.

- Behavioral assays after stress exposure or viral-mediated manipulations do not cover all symptomatology observed in individuals with MDD. Anhedonia for example is generally assessed using sucrose preference and social avoidance with a social interaction test. Addition of these features would bonify the present study.

- Similarly, lack of behavioral effects when viral-mediated manipulations to overexpress vegfc are performed in male mice subjected to SCVS would greatly strengthen the sex-specific conclusions.
- Uneven distribution of intracisternally-injected tracer with enrichment in the mPFC and VTA is very intriguing. Unfortunately, it is unclear right now if experiments were performed in a naïve female mouse and if yes, is exposure to SCVS altering distribution? Is this uneven distribution sex-specific?
- Panels of Fig.3d-e could be grouped and statistically analyzed with b-c to confirm that viral-mediated manipulation can reverse completely astrocytic changes. Same comment for h-i/f-g. At the moment it is hard to make conclusions with normalization performed on separate graphs and thus, either unstressed controls or mice injected with only eGFP (different y-axis).
- Data provided to suggest that viral-mediated manipulation impairing meningeal lymphatic structural and function in male mice can induce development of a depression-like phenotype do not support this conclusion. Indeed, when comparing behavioral data provided in Supp.Fig.2, behavioral effects appear to be driven by the control virus with values similar for mice injected with VEGFF3d1-4+SCVS (Fig.4k-l) and unstressed controls in the ST (about 100 sec) and FST (125-130 sec). Additional tests are needed.
- How can modest behavioral effects observed in male mice following viral-mediated manipulation be reconciled with drastic changes in the male mPFC astrocyte morphology and VTA neuronal activation? Is it possible that distribution of an intracisternally-injected tracer would be different in males and thus, targeted regions are not appropriate here?
- Transcriptomic profiling of meningeal lymphatic endothelial cells of female mice exposed or not to SCVS is highly relevant. The comparison with DEGs identified in a study related to aging are intriguing but why not explore gene lists associated with human depression? Even in the context of heterogeneous preparation confirmation of gene expression changes in human tissue databases, and identified here in stressed female mice, would be more appropriate.

Minor:

- It is unclear in the abstract which brain regions were characterized for brain drainage function. Same comment for injections of the AAVs. VTA and mPFC are mentioned at the end but it appears disconnected.
- Timeline of SCVS vs tissue collection should be justified. Why wait until day 11? This is unconventional with most studies collecting tissue the day after the last stress exposure session vs 72h here. Were other time points (24h, 48h) analyzed for Lyve1 expression with no difference observed?
- Mouse cohorts that went through behavioral characterization (Supp. Fig.1-2) are twice larger for female vs male mice, but it is unclear why. It is surprising that power analysis gave n=20-25 for females vs only n=9 for males. Comparable number of animals for each sex may render SCVS-induced behavioral effects in males significant.
- In the Reporting Summary, "No data were excluded" is mentioned in Life sciences study design, however, the number of mice for behavioral tests differs between graphs (ex: Supplementary Fig 1 and 2, Fig 1, Fig 2, Fig 4). Please explain in detail how outliers were determined.
- Were proper statistic tests conducted to assume normality of the datasets (Kolmogorov–Smirnov, Shapiro–Wilk test, etc.)?
- The Data availability statement is insufficient and no access to gene lists produced by RNA sequencing in suitable repositories was provided.

Response to reviewers

Manuscript: NCOMMS-21-44233A

Dai et al. A functional role of meningeal lymphatics in sex difference of stress susceptibility in mice

We are pleased to submit our revised manuscript now entitled “A functional role of meningeal lymphatics in sex difference of stress susceptibility in mice” for consideration for publication in *Nature Communication*.

We thank the reviewers for considering our work very exciting, of great potential significance, intriguing and novel. We are also grateful for the valuable critiques and suggestions raised by the reviewers, which really help us clarify and strengthen our manuscript. We have addressed all the reviewers’ comments, by performing significant amount of additional experiments as well as revision of the text. We feel that the manuscript is much improved by virtue of reviewers’ suggestions.

Please find below our point-by-point responses to address the critiques and comments made by the reviewers.

Reviewer #1

In this study, Dai et al explored whether meningeal lymphatic dysfunction contributes to stress induced depression. Using a sub-chronic variable stress (SCVS) paradigm to induce depressive-like behaviors, they show here that SCVS leads to impaired meningeal lymphatic drainage and altered lymphatic vasculature morphology in female but not male mice and that this correlates with the female-biased induction of depression-related behaviors in this SCVS model. To interrogate a mechanistic role for defective meningeal lymphatic drainage in the depressive phenotypes that develop in SCVS treated female mice, they showed that VEGFC-induced boosting of meningeal lymphatic drainage rescues the depressive phenotypes in female SCVS mice. Moreover, they further demonstrate that pre-existing meningeal lymphatic defects that are brought on with AAV-VEGFR3d1-4 pretreatment in male mice promotes the development of depression-associated behaviors following SCVS treatment. Interestingly, they also report that SCVS conditions lead to diminished activity of dopaminergic neurons in the ventral tegmental area and that this could be ameliorated with VEGFC-mediated boosting of meningeal lymphatic function. Overall, I am quite enthusiastic about this work and found their findings to be very exciting and of great potential significance. Despite these strengths, a few issues and technical shortcomings were noted, which I believe need to be addressed in order to solidify their central findings and conclusions.

Response:

We thank the reviewer for the valuable comments and suggestions. We are pleased that the reviewer is enthusiastic about our work. We have addressed all the issues raised by the reviewer.

Major comments:

1. A few important controls are missing throughout the manuscript. For instance, it will be important to show that meningeal lymphatic drainage is not altered in SCVS treated male mice in comparison to non-stressed male controls (similar to the studies done in SCVS females in Figure 1n). It will also be important to evaluate if VEGFC treatment alone in non-stressed female mice impacts any of the behaviors presented in Figure 2l-r. Likewise, it is also important to provide non-stressed male controls for the behavior studies exploring the effects of lymphatic ablation (AAV-VEGFR3_{d1-4} treatment) in Figure 4k-r.

Response:

We have performed the following control experiments as suggested by the reviewer:

(1) We have included additional experimental data to validate the lack of impairment in meningeal lymphatics (mLV) of male mice after sub-chronic variable stress (SCVS). Using immunofluorescence staining and intracisternal tracer injection, we showed that the intensity of immunofluorescence staining for LYVE1, the coverage area of LYVE1-labelled mLV as well as the diameters of mLV in the superior sagittal sinus (SSS), the transverse sinus (TS) and the confluence of sinuses (COS) of the dura mater remained unchanged after SCVS in male mice. In line with the lack of structural deficits of meningeal lymphatics, the amount of intracisternally-injected 70kD Dextran-Texas Red tracer detected in SSS, TS and COS areas of the dura mater as well as in the deep cervical lymph nodes (dCLNs) were unaltered by SCVS in male mice. These data are included in the **new Figure 1 (e, k-o, r-s)**.

(2) We have included additional experimental data, which showed that intracisternal injection of AAV-VEGFC to non-stressed female mice promoted meningeal lymphatic growth, as evident by increased intensity of LYVE1 immunofluorescence staining, the coverage area of LYVE1-labelled mLV as well as the diameters of mLV in the SSS, TS and COS of the dura mater. However, we found that the VEGFC treatment alone in non-stressed female mice did not affect animals' performance in the splash test, the forced swim test, the novelty suppressed feeding test and the open field test. Furthermore, the VEGFC treatment alone did not alter astrocytic protein expression (including S100 β and GFAP) in the medial prefrontal cortex (mPFC) and cFOS expression of ventral tegmental area (VTA) (the **new Supplementary Figure 5**). These new data, together with the data presented in Figure 2, suggest that enhancement of meningeal lymphatics *per se* was not sufficient to alter depression- or anxiety-like behaviors in female mice, nor alter the mPFC astrocytes and the VTA dopaminergic neuronal activity, but rather, increased their resilience to stress.

(3) Furthermore, we have performed experiments to examine whether VEGFR3_{d1-4} treatment alone affected the depression- and anxiety-like behaviors of non-stressed male mice. As pointed out by the reviewer 2, a more appropriate control for VEGFR3_{d1-4} shall be VEGFR3_{d4-7}, which is derived from the VEGFR3 extracellular portion but does not bind to VEGFC (Alitalo et al. 2013; Antila et al. 2017). Therefore, for this experiment, the male mice in the control group were intracisternally injected with AAV that expressed VEGFR3_{d4-7}, rather than eGFP, as shown in our previous version of the manuscript. The results revealed that the VEGFR3_{d1-4} treatment alone did not affect the performance of non-stressed male mice in the tests for depression-like behaviors, including the splash test, the forced swim test and the novelty suppressed feeding test. There was a trend towards decrease in the time spent in the center zone of the open field arena ($p = 0.055$) for the VEGFR3_{d1-4} treatment group, indicating a potential anxiogenic phenotype (the **new Supplementary Figure 10a-h**).

Together, the newly added data suggest that (1) SCVS impaired the mLV of female but not male mice, and (2) manipulation of meningeal lymphatics *per se* is not sufficient to alter depression-like behaviors in male and female mice, but rather, changes their stress susceptibility.

References

1. Alitalo AK, et al. VEGF-C and VEGF-D blockade inhibits inflammatory skin carcinogenesis. *Cancer Res* 73, 4212-4221 (2013).
2. Antila S, et al. Development and plasticity of meningeal lymphatic vessels. *J Exp Med* 214, 3645-3667 (2017).

2. It is unclear based on the data provided in Figure 3 whether CSF influx and ISF diffusion (glymphatic function) are altered in their SCVS model. In Figure 3a, they appear to show glymphatic drainage of OVA tracer in the brain parenchyma. However, they do not go on to investigate whether this is affected by SCVS or if this can be modulated by VEGFC treatment. These specific studies should be conducted to address this issue and to also support their suggestive astrocyte end-feet data presented in Figure 3. Importantly, they should also provide quantification of the OVA tracer glymphatic drainage in addition to the representative images presented in Figure 3a.

Response:

The presentative images on the distribution of intracisternally-injected ovalbumin-Alexa Fluor 647 (OVA) tracer in brain parenchyma provided in the original Figure 3a were from naïve female mice. We have now extended this study by adding the following experiments and analyses:

(1) We have added the quantification of the OVA tracer across multiple brain regions at different time points (15 min, 30 min and 60 min) post-intracisternal injection in the naïve female mice. As shown in the representative images and the quantification graph, the distribution of intracisternally-injected tracer was uneven in the brain parenchyma, with an enrichment in the mPFC at 15-30 min post-injection, and in the hypothalamus and the VTA at 30-60 min post-injection (**the revised Figure 3a-b**).

(2) We have performed experiments to examine whether the distribution of intracisternally-injected OVA tracer in the brain parenchyma is affected by SCVS and intracisternal-injection of AAV-VEGFC to female mice. For this experiment, we chose the 1 h post-injection time point, the same time point at which we found reduction of the tracer arrived at dura mater and dCLNs of female mice after SCVS (Figure 1). After SCVS, significantly less tracer was found in the brain parenchyma, which however was prevented by the VEGFC treatment that improved meningeal lymphatics (**the new Supplementary Figure 7a-c**). These findings are consistent with previous reports that alteration in meningeal lymphatics could lead to changes in the rate of CSF influx and ISF diffusion in the brain parenchyma (Louveau et al. 2017; Da Mesquita et al. 2018). Therefore, our data suggest that CSF influx and the ISF diffusion in female mice is impaired by SCVS, which however can be prevented by improvement of meningeal lymphatics.

References

1. Louveau A, Plog BA, Antila S, Alitalo K, Nedergaard M, Kipnis J. Understanding the functions and relationships of the glymphatic system and meningeal lymphatics. *J Clin Invest* 127, 3210-3219 (2017).
2. Da Mesquita S, et al. Functional aspects of meningeal lymphatics in ageing and Alzheimer's disease. *Nature* 560, 185-191 (2018).

Minor comments:

1. In line 93, it should be “interleukin-6” and not “interleukine-6”.

Response:

We have corrected this typo (main text, page 4, line 92).

2. In line 94, I think it should be “reverse fashion” and not “reversed fashion”.

Response:

We have corrected this typo (main text, page 5, line 93).

Reviewer #2

In this manuscript, the authors are describing a role of meningeal lymphatic vasculature in a model of sub-chronic variable stress that only leads to depressive-like behavior in females. Experimental data using this sex biased stress model in combination with gain and loss of function manipulations of meningeal lymphatic vasculature suggest that modulation of meningeal lymphatic drainage can be considered a therapeutic target in depression. However, all the data is correlative and does not imply a direct involvement of the meningeal lymphatic system, or a mechanism through which reduced (or enhanced) meningeal lymphatic drainage is underlying the described phenomena in the context of stress-induced depressive-like behavior. There are several concerns that need to be addressed regarding some of the described concepts, the current provided data, and data interpretation. There are also additional experiments and data that need to be provided to allow the authors to fully substantiate their claims/conclusions.

Response:

We thank the reviewer for the helpful critiques and suggestions, which are of great value to help strengthen our manuscript. We have obtained additional experimental data and revised the manuscripts to address the concerns raised by the reviewer.

Specific comments/concerns/experimental requests:

1. The authors refer (at least twice) that the meningeal lymphatics drain “excess” fluid, macromolecules and immune cells. This is a wrong description of the process. Lymphatic vessels in the meninges constantly, and rather unselectively, drain CSF molecules that reach the dural stroma. Molecular content from the CSF is constantly drained into the cervical lymph nodes – not only in situations where molecules are “in excess”. The process of immune cell drainage is quite different. This is not a passive, but active process, that depends on chemokine gradients alongside other mechanisms. The authors need to be thorough when describing these processes.

Response:

We thank the reviewer for pointing out the misused word “excess” in describing the meningeal lymphatic drainage process. We have removed the word “excess” in the abstract. We have also modified the related

sentences in the Introduction to “These meningeal lymphatics constantly drain fluid and macromolecules from the CNS to the periphery by connecting to the deep cervical lymph nodes (dCLN), as well as play an important role in the active transportation of immune cells” (main text, page 3, line 49; page 5, line 101-103).

2. Measurement of Lyve1 mRNA in the dura matter (Fig. 1a and b) cannot be used as a surrogate of lymphatic vasculature function. There is a very well described population of resident dural macrophages that express high levels of Lyve1 mRNA and LYVE-1 protein.

Response:

We have included additional experimental data to validate the lack of impairment in meningeal lymphatics (mLV) of male mice after sub-chronic variable stress (SCVS). Using immunofluorescence staining and intracisternal tracer injection, we showed that the intensity of immunofluorescence staining for LYVE1, the coverage area of LYVE1-labelled mLV as well as the diameters of mLV in the superior sagittal sinus (SSS), the transverse sinus (TS) and the confluence of sinuses (COS) of the dura mater remained unchanged after SCVS in male mice. In line with the lack of structural deficits of meningeal lymphatics, the amount of intracisternally-injected 70kD Dextran-Texas Red tracer detected in SSS, TS and COS areas of the dura mater as well as in the deep cervical lymph nodes (dCLNs) were unaltered by SCVS in male mice. These data are included in the **new Figure 1 (e, k-o, r-s)**.

3. What are the differences between basal adult male and female meningeal lymphatic vessel morphology and drainage capacity to the deep CLNs? This analysis should be performed.

Response:

We have included new experimental data, which revealed that there was no significant difference in the morphology of meningeal lymphatic vessels comparing non-stressed adult female and male mice in several measures, including the intensity of LYVE1 immunofluorescence staining, the coverage area of meningeal lymphatic vessels as well as the diameters of LYVE1-labelled meningeal lymphatic vessels in the SSS, TS and COS areas of the dura mater. Furthermore, similar amount of intracisternally-injected 70kD Dextran-Texas Red tracer was detected in the SSS, TS and COS areas of the dura mater as well as in the dCLNs of non-stressed female versus male mice, suggesting that the brain drainage capacity to the dCLNs of basal female and male mouse brain was also similar (the **new Supplementary Figure 3d-h**). Collectively, these results suggest that, consistent with previous reports (Da Mesquita et al. 2018; Ahn et al. 2019), basal adult female and male meningeal lymphatic vessels were similar in structure and drainage capacity.

References

1. Da Mesquita S, et al. Functional aspects of meningeal lymphatics in ageing and Alzheimer's disease. *Nature* 560, 185-191 (2018).
2. Ahn JH, et al. Meningeal lymphatic vessels at the skull base drain cerebrospinal fluid. *Nature* 572, 62-66 (2019).

4. Why are females, but not males, affected using this stress protocol? Authors have to provide more data about this. One suggestion is to measure the levels of corticosterone during day and night in

females and males from control and SCVS groups. This will provide additional clues on why meningeal lymphatics might be affected in females but not males.

Response:

In order to address the reviewer's question and to examine whether the different levels of corticosterone may be responsible for SCVS-induced changes in meningeal lymphatics of female but not male mice, we measured the serum corticosterone (CORT) levels before and after the completion of SCVS in both female and male mice. Due to the dynamic nature of serum CORT levels in the morning and at night as mentioned by the reviewer and reported previously (Dumbell et al. 2016), we measured the serum CORT levels at 8:00 AM and 8:00 PM. Our results showed that serum CORT levels were higher at 8:00 PM as compared with those at 8:00 AM in the naïve female and male mice without detectable sex difference. Notably, when comparing serum CORT levels in the morning (8:00 AM), increases in CORT levels were evident in both female and male mice after SCVS as compared to the naïve mice. However, dramatic decreases in CORT levels were observed in female and male mice after SCVS when the serum CORT levels were examined at night (8:00 PM). Therefore, the circadian oscillation of serum CORT levels was blunted after SCVS, with similar magnitudes of changes detected in both female and male mice (the **new Supplementary Figure 11a-c**).

To directly examine the effect of CORT on meningeal lymphatics, male mice received consecutive CORT injection between 9-11 AM daily for 28 days, which generated depression-like phenotypes as reported previously (Gobinath et al. 2018). However, such chronic CORT administration did not induce detectable changes in the meningeal lymphatics as measured by the LYVE1 expression, the coverage of LYVE1-labelled meningeal lymphatics as well as the diameter of meningeal lymphatics (the **new Supplementary Figure 11d-k**). Our findings therefore suggested that corticosterone was unlikely to be responsible for the SCVS-induced sex different impairment of meningeal lymphatics.

References

1. Dumbell R, Matveeva O, Oster H. Circadian Clocks, Stress, and Immunity. *Front Endocrinol (Lausanne)* 7, 37 (2016).
2. Gobinath AR, et al. Voluntary running influences the efficacy of fluoxetine in a model of postpartum depression. *Neuropharmacology* 128, 106-118 (2018).

5. Are different levels of corticosterone in stressed females and males responsible for the different phenotypes in terms of meningeal lymphatic function. What is the mechanism underlying the loss of meningeal lymphatic coverage in stressed females?

Response:

As addressed in the previous question raised by the reviewer, we have conducted new experiments to show that the corticosterone was unlikely to contribute to the sex difference in SCVS-induced meningeal lymphatic impairment.

To identify the potential regulatory mechanism that is involved in the SCVS-induced loss of meningeal lymphatics coverage in the female mice, we turn back to the RNAseq data to look for the differentially expressed genes induced by SCVS in the meningeal lymphatic endothelial cells of female mice. A robust induction of chemokine (C-C motif) ligand 6 (CCL6) by SCVS was identified as shown in the volcano plot of

gene expression changes by comparing the naïve group versus the SCVS group (Figure 7h). To examine the potential regulatory role of CCL6 in SCVS-induced meningeal lymphatic impairment, we have performed an additional functional manipulation experiment by intracisternal infusion of CCL6-neutralizing antibody to disrupt CCL6 signaling (Zhang et al. 2018). The results showed that functional blockage of CCL6 prevented SCVS-induced depression-like phenotypes and impairment of meningeal lymphatics in female mice (the **new Figure 8**). CCL6 has been previously shown to play critical roles in the pathogenesis of IL-13-induced tissue inflammation and the homeostasis of hematopoietic stem cells (Ma et al. 2004; Zhang et al. 2018). However, no studies have yet reported a role of CCL6 in regulating the meningeal lymphatics and depression-like phenotypes. Our findings therefore provided a possible mechanism underlying the SCVS-induced reduction of meningeal lymphatic coverage in the female mice.

It is also worth noting that the CCL6 ortholog gene product in human, CCL23, has been reported as a diagnostic biomarker in the serum or cerebrospinal fluid of patients with Alzheimer's disease and ischemic stroke (Simats et al. 2018; Faura et al. 2020). Whether human CCL23 may be involved in the regulation of meningeal lymphatics and the disease progression of neurodegenerative disorders and depression is warranted for further studies. We have included the new data and discussion in the revised version of the manuscript (**main text, page 21-22, line 497-511**).

References

1. Zhang C, et al. Eosinophil-derived CCL-6 impairs hematopoietic stem cell homeostasis. *Cell Res* 28, 323-335 (2018).
2. Ma B, Zhu Z, Homer RJ, Gerard C, Strieter R, Elias JA. The C10/CCL6 chemokine and CCR1 play critical roles in the pathogenesis of IL-13-induced inflammation and remodeling. *J Immunol* 172, 1872-1881 (2004).
3. Faura J, et al. CCL23: A Chemokine Associated with Progression from Mild Cognitive Impairment to Alzheimer's Disease. *J Alzheimers Dis* 73, 1585-1595 (2020).
4. Simats A, et al. CCL23: a new CC chemokine involved in human brain damage. *J Intern Med* 283, 461-475 (2018).

6. The results in Fig. 1g and I suggest that less CSF is reaching the vicinity of the meningeal lymphatics, which means that the deficits in the amount of tracer that is reaching the dCLN is not due to impaired drainage by meningeal lymphatics, but by a potential impairment in the pathway that connects the brain perivascular/subarachnoid spaces (that become filled with the injected tracer) and the dural stroma. Similar results are also observed in the data in Fig. 4c and h, where stressed female mice treated with VEGFR3-(1-4) also show less fluorescent tracer (hence less CSF) reaching the dural compartment and the draining lymphatic vessels. The opposite is observed in Fig. 2c and h, where VEGF-C leads to more fluorescent tracer reaching the dural tissue. Most of the fluorescent tracer that reaches the dura in postmortem tissue is present inside dural macrophages (Rustenhoven et al, *Cell* 2021). Another possible explanation is that the dural macrophages in the stressed female present deficits in phagocytic capacity. In sum, these observations put into question the loss of drainage capacity by meningeal lymphatics in the context of SCVS. The authors need to complement the measurements of fluorescent tracer in the dura with measurements of fluorescent tracer left in the CSF (collected from the cisterna magna or the lateral ventricle) at the time of euthanasia (1 h post icm injection). It is also very important to perform experiments to assess the activation status of dural

myeloid cells (e.g., macrophages), by flow cytometry or other techniques, to discard (or consider) the involvement of macrophages (and their secreted cytokines) in the described phenomena.

Response:

We agree with the reviewer on the point that our data suggest less CSF reaching the vicinity of the meningeal lymphatics after SCVS in female mice, or after SCVS in male mice intracisternally injected with AAV-VEGFR3_{d1-4}, whereas the opposite occurred in female mice intracisternally injected with AAV-VEGFC and subjected to SCVS. There are three potential mechanisms underlying such alternation in CSF drainage: (1) Since the coverage area and the diameter of meningeal lymphatics were altered by SCVS in female mice and manipulation of the VEGFC-VEGFR3 signaling, we suspected that these morphological changes might result in functional alterations of the drainage capacity of meningeal lymphatics, which in turn alter the rate of CSF flow inside the brain due to the continuous drainage of CSF from the brain to the meningeal lymphatics (Louveau et al. 2017; Da Mesquita et al. 2018). However, we cannot exclude the possibility that (2) alterations in the pathway that connects the brain perivascular/subarachnoid spaces and the dura stroma occurred, or (3) alternation in the phagocytic capacity of macrophages along the dura transverse sinus (TS). We have revised the manuscript text to remove the claim that the drainage capacity of the meningeal lymphatics *per se* was altered, but rather, to describe the phenomena as alterations in the general brain drainage function, and add in the discussion of potential mechanisms (**main text, page 20-21, line 474-485**).

To address the reviewer's concern and to provide further understanding of regulation of CSF drainage by SCVS in female mice, we have performed the following additional experiments:

(1) To examine whether CSF influx and ISF diffusion in the brain parenchyma are affected by SCVS, we injected OVA-Alexa Fluor 647 tracer into the cisterna magna of naïve female mice and those experienced SCVS. The mice were perfused 1 h post-injection. After SCVS, significantly less tracer was found in the brain parenchyma. However, intracisternal infusion of AAV-VEGFC that improved meningeal lymphatics prevented the SCVS-induced reduction of tracer in the brain parenchyma (the **new Supplementary Figure 7a-c**). These findings are consistent with previous reports that alteration in meningeal lymphatics could lead to changes in the rate of CSF influx and ISF diffusion in the brain parenchyma (Louveau et al. 2017; Da Mesquita et al. 2018). Therefore, our data suggest that CSF influx and the ISF diffusion in female mice is impaired by SCVS, which however can be prevented by improvement of meningeal lymphatics.

(2) To examine the intracisternally-injected tracer left in the CSF, we have tried collecting CSF from the cisterna magna or lateral ventricle after intracisternal tracer injection. However, due to technical issues that have yet to be resolved, we failed to get consistent and reliable results. Therefore, we had to take an indirect approach. For this experiment, we injected Evans Blue tracer into the cisterna magna of naïve female mice and those experienced SCVS. The mice were sacrificed 1 h later and their whole brains were collected and homogenized. The concentration of tracer in the homogenate was measured by a plate-reader to estimate the amount of tracer left in the brain parenchyma together with that in the brain ventricles. The results showed that there was no difference between the naïve mice and those experienced SCVS (the **new Supplementary Figure 7d-e**). Since intracisternally-injected tracer detected in the brain parenchyma, in the vicinity of meningeal lymphatics and in the dCLNs was all decreased after SCVS (Figure 1 and the new Supplementary Figure 7a-c), these newly added data indicate a possibility of increased accumulation of CSF in the brain ventricles and the subarachnoid space after SCVS.

(3) To evaluate whether the phagocytic activity of macrophages near meningeal lymphatics was altered by SCVS, we used CX3CR1-eGFP transgenic mice to label macrophages with eGFP (Louveau et al. 2018). The results showed that after SCVS, less tracer was detected in the TS of the dura mater. Also, there was significantly less tracer detected in the eGFP-labelled macrophages near the meningeal lymphatics in the TS area. However, the percentage of tracer detected within the macrophages versus total tracer in the TS of the dura mater was not significantly altered after SCVS (the **new Supplementary Figure 7h-k**). These data indicate that the phagocytic activity of macrophages was not significantly altered by SCVS.

(4) Impairment of CSF influx is often associated with impairment in the efflux of ISF macromolecules from the brain parenchyma (Iliff et al. 2012; Da Mesquita et al. 2018). To examine whether SCVS impairs the ISF efflux function in the brain parenchyma, we injected both 70kD Dextran-Texas Red tracer and 40kD Dextran-Alexa Fluor 488 tracer into the mPFC and the VTA. Indeed, we found that for both brain regions, the efflux of intracerebrally injected tracer was reduced after SCVS (the **new Supplementary Figure 7l-q**).

Collectively, these data suggest that SCVS resulted in a reduction of the influx and diffusion of CSF macromolecules in the brain parenchyma, which is likely affected by impairment of meningeal lymphatic vessels, and was accompanied by a reduction in the efflux of ISF macromolecules. The alteration in CSF flow rate might lead to a reduction of CSF reaching the vicinity of meningeal lymphatics.

References

1. Louveau A, Plog BA, Antila S, Alitalo K, Nedergaard M, Kipnis J. Understanding the functions and relationships of the glymphatic system and meningeal lymphatics. *J Clin Invest* 127, 3210-3219 (2017).
2. Da Mesquita S, et al. Functional aspects of meningeal lymphatics in ageing and Alzheimer's disease. *Nature* 560, 185-191 (2018).
3. Louveau A, et al. CNS lymphatic drainage and neuroinflammation are regulated by meningeal lymphatic vasculature. *Nat Neurosci.* 21(10):1380-1391 (2018)
4. Iliff JJ, et al. A paravascular pathway facilitates CSF flow through the brain parenchyma and the clearance of interstitial solutes, including amyloid beta. *Sci Transl Med* 4, 147ra111 (2012).

7. The control for the VEGFR3-(1-4) viral vector should be injection of an AAV expressing the domains 4-7 of the receptor (coupled to a Ig or Fc domain), and not eGFP. It is not clear where the authors manufactured or purchased the AAV constructs. Are they commercially available? If this is the case, they need to provide the references. If this is not the case, more details must be provided on how the viral constructs were generated. This is mandatory to allow reproducibility of the experimental results.

Response:

We agree with the reviewer that the more suitable control for the experiment using VEGFR3_{d1-4} viral vector to sequester VEGFC should be a viral vector that express the domain 4-7 of VEGFR3, which does not bind to VEGFC. In the revised manuscript, we have constructed the new control AAV expressing domain 4-7 of VEGFR3 fused with a mouse Fc domain (AAV-CAG-VEGFR3_{d4-7}-Fc) and repeated the experiments as suggested by the reviewer. The results were shown in the **revised Figure 4**, the **new Figure 5c-j** and the **new Supplementary Figure 6f-j**, in which male mice that received intracisternal injection of AAV overexpressing VEGFR3_{d1-4}, compared to those injected with AAV overexpressing VEGFR3_{d4-7}, showed impaired meningeal

lymphatics, drainage of CSF tracer, depression-like behaviors as well as abnormalities in the mPFC and the VTA after SCVS.

For AAV viral vectors used in this study, the AAVs expressing eGFP (AAV-CMV-eGFP-3FLAG, AAV-CAG-eGFP), VEGFC (AAV-CMV-VEGFC-P2A-eGFP-3FLAG), and two VEGFR3 mutants (AAV-CAG-VEGFR3d1-4-IgG Fc and AAV-CAG-VEGFR3d4-7-IgG Fc) were custom-made by OBiO (Shanghai, China). Briefly, PCR-amplified fragments of mouse *Vegfc* coding sequence (NM_009506.2) were cloned into the pAAV-CMV-eGFP-3FLAG backbone vector (CMV: cytomegalovirus promoter; eGFP: eGFP, enhanced green fluorescent protein; 3FLAG: 3 copies of FLAG epitope tag) with self-cleaving 2A peptide (P2A) in between sequences encoding VEGFC and eGFP (AAV-CMV-VEGFC-P2A-eGFP-3FLAG) 30, 66. PCR-amplified fragments of mouse VEGFR3-coding sequence (NM_008029.3) encoding amino acid sequence 45–415 of mouse VEGFR3 domain 1-4 (VEGFR3d1-4) and amino acid sequence 331-764 of VEGFR3 domain 4-7 (VEGFR3d4-7) were cloned into the pAAV-CAG-IgG Fc-HA backbone vector and fused in frame with the mouse IgG Fc domain (CAG: the CMV early enhancer/chicken beta-actin promoter; IgG: mouse IgG Fc domain; HA: the hemagglutinin epitope tag) for constructing the pAAV-CAG-VEGFR3d1-4-IgG Fc and pAAV-CAG-VEGFR3d4-7-IgG Fc plasmids, as reported previously 39, 92. The plasmids of AAV-CMV-eGFP-3FLAG and AAV-CMV-VEGFC-P2A-eGFP-3FLAG were packaged into AAV serotype 1 (AAV1). The plasmids of AAV-CAG-VEGFR3d1-4-IgG Fc, AAV-CAG-VEGFR3d4-7-IgG Fc, and AAV-CAG-eGFP were packed into AAV serotype 9 (AAV9). We have also included these descriptions in the Methods section of the revised manuscript (**main text, page 30-31, line 700-718**).

References

1. Da Mesquita S, et al. Functional aspects of meningeal lymphatics in ageing and Alzheimer's disease. *Nature* 560, 185-191 (2018).
2. Leppanen VM, et al. Structural and mechanistic insights into VEGF receptor 3 ligand binding and activation. *Proc Natl Acad Sci U S A* 110, 12960-12965 (2013).
3. Antila S, et al. Development and plasticity of meningeal lymphatic vessels. *J Exp Med* 214, 3645-3667 (2017).

8. What is the effect of loss of meningeal lymphatic function (upon VEGFR3-(1-4)) or gain of meningeal lymphatic function (upon VEGF-C) on the performance of female and male (without the influence of stress) mice in the ST, FST, NSF? It is possible that defective meningeal lymphatic drainage per se is enough to cause depressive-like behavior in males and/or females. To my knowledge, this has never been explored. These controls are missing in all the data presented in Figs 2 and 4. For the sake of the overall message of the study, it is imperative that these experiments are performed, and this data is included in the manuscript.

Response:

We have included additional experimental data, which showed that intracisternal injection of AAV-VEGFC to non-stressed female mice promoted meningeal lymphatic growth, as evident by increased intensity of LYVE1 immunofluorescence staining, the coverage area of LYVE1-labelled mLV as well as the diameters of mLV in the SSS, TS and COS of the dura mater. However, we found that the VEGFC treatment alone in non-stressed female mice did not affect animals' performance in the splash test, the forced swim test, the novelty

suppressed feeding test and the open field test. Furthermore, the VEGFC treatment alone did not alter astrocytic protein expression (including S100 β and GFAP) in the medial prefrontal cortex (mPFC) and cFOS expression of ventral tegmental area (VTA) (the **new Supplementary Figure 5**). These new data, together with the data presented in Figure 2, suggest that enhancement of meningeal lymphatics *per se* was not sufficient to alter depression- or anxiety-like behaviors in female mice, nor alter the mPFC astrocytes and the VTA dopaminergic neuronal activity, but rather, increased their resilience to stress.

Furthermore, we have performed experiments to examine whether VEGFR3_{d1-4} treatment alone affected the depression- and anxiety-like behaviors of non-stressed male mice. As pointed out by the reviewer, a more appropriate control for VEGFR3_{d1-4} shall be VEGFR3_{d4-7}. For this experiment, the male mice in the control group were intracisternally injected with AAV that expressed VEGFR3_{d4-7}. The results revealed that the VEGFR3_{d1-4} treatment alone did not affect the performance of non-stressed male mice in the tests for depression-like behaviors, including the splash test, the forced swim test and the novelty suppressed feeding test. There was a trend towards decrease in the time spent in the center zone of the open field arena ($p = 0.055$) for the VEGFR3_{d1-4} treatment group, indicating a potential anxiogenic phenotype (the **new Supplementary Figure 10a-h**).

Together, the newly added data suggest that manipulation of meningeal lymphatics *per se* is not sufficient to alter depression-like behaviors in male and female mice, but rather, changes their stress susceptibility.

9. Treatment of stressed female with VEGF-C seems to result in better meningeal lymphatic drainage, alongside with more S100b+ and GFAP+ cells in the mPFC and c-FOS+ cells in the VTA. These observations are correlated but have not been functionally linked by the data provided by the authors. VEGF-C has the capacity to signal on other cells in the brain that express VEGFR2 or VEGFR3, hence the observed effects might well be lymphatic-independent. The authors need to provide experimental evidence that points to a potential mechanism linking increased lymphatic drainage, due to VEGF-C, to the described region-specific alterations in glial and neuronal responses.

Response:

Although VEGFC has the capacity to signal on other cells in the brain that express VEGFR2 or VEGFR3, after intracisternal injection, we found AAV-infected cells that expressed eGFP were mostly located in the transverse sinus (TS), the confluence of sinuses (COS) and part of the superior sagittal sinus (SSS) that covering the olfactory bulb, and near LYVE1-labelled meningeal lymphatic vessels. We did not observe infected cells in the brain parenchyma, except occasionally few cells in the cerebellum that near injection site. Of note, similar restricted distribution of intracisternally injected AAV-VEGFC was also noted in a previous study, which reported that this viral approach did not affect meningeal blood vessel coverage or proliferation of neural stem cells in the hippocampus, which could potentially be mediated by VEGFC signaling (Da Mesquita et al. 2018).

To avoid the potential lymphatic-independent effects of manipulation of VEGFC-VEGFR3 signaling on stress susceptibility, we have performed additional experiment in which the lymphatic vessels afferents to dCLNs were lighted to block the drainage pathway through meningeal lymphatics. As shown in the **new Figure 6**, male mice that received the ligation surgery, compared to those received sham operation, showed impaired CSF drainage to the dCLN, depression-like behaviors, as well as abnormalities in the mPFC and the VTA after

SCVS. These findings support the notion that impairment in the drainage through meningeal lymphatics could increase susceptibility to stress.

To gain further understanding of potential mechanism linking increased lymphatic drainage, due to VEGFC, to the described region-specific alterations in glial and neuronal responses, we performed lipidomic analyses of the mPFC, the VTA and the dura mater, comparing naïve female mice injected with AAV-eGFP, female mice injected with AAV-eGFP and experienced SCVS, and those injected with AAV-VEGFC and experienced SCVS. One of the key physiological functions of lymphatic vessels is to absorb and transport lipids (Oliver et al. 2020). On the other hand, lipids are enriched and engaged in a wide range of physiological functions in the brain (Ralhan et al. 2021). It has been reported that chronic unpredictable stress alters lipid profiles in the mPFC (Oliveira et al. 2016), and that the activity of VTA dopaminergic neurons can be modulated by triacylglycerols (TAGs). As mentioned in the response to previous question by the reviewer, we found that after injection into the mPFC or the VTA, the efflux of tracer was significantly reduced after SCVS. We therefore wondered whether SCVS and the increased meningeal lymphatic drainage due to VEGFC could regulate lipid accumulation in the mPFC and the VTA. The results from the lipidomic analyses showed that after SCVS, significant increase in a variety of lipids including long-chain TAGs was notable in the mPFC while a small subset of TAGs were also increased in the VTA after SCVS, which were partially prevented by the VEGFC treatment. Importantly, the VEGFC treatment significantly increased a panel of lipid molecules, including long-chain TAGs, in the dura mater. These findings were consistent with the notion that increased drainage through meningeal lymphatic by the VEGFC treatment might facilitate lipid efflux and reduced stress-induced accumulation of lipid molecules in the mPFC and the VTA of female mice (the **new Supplementary Figure 9a-e**).

To examine the functional link between identified lipids significantly changed by SCVS in the mPFC and the VTA of female mice, we focused on the long-chain TAGs, which were accumulated in both brain regions after SCVS and the accumulation were partially prevented by the VEGFC treatment. The results showed that direct injection of TAGs into the mPFC led to significant loss of astrocytes, whereas in the VTA the injection decreased the expression of c-FOS in the TH+ dopaminergic neurons (the **new Supplementary Figure 9f-i**). Collectively, these findings suggest that enhancement of meningeal lymphatics by VEGFC prevented SCVS-induced abnormalities in the mPFC astrocytes and the VTA dopaminergic neurons in the female mice, likely through the facilitation of long-chain TAG efflux from the two brain regions.

References

1. Da Mesquita S, et al. Functional aspects of meningeal lymphatics in ageing and Alzheimer's disease. *Nature* 560, 185-191 (2018).
2. Oliver G, Kipnis J, Randolph GJ, Harvey NL. The Lymphatic Vasculature in the 21(st) Century: Novel Functional Roles in Homeostasis and Disease. *Cell* 182, 270-296 (2020).
3. Ralhan I, Chang CL, Lippincott-Schwartz J, Ioannou MS. Lipid droplets in the nervous system. *J Cell Biol* 220(7): e202102136 (2021).
4. Oliveira TG, et al. The impact of chronic stress on the rat brain lipidome. *Mol Psychiatry* 21, 80-88 (2016).
5. Berland C, et al. Circulating Triglycerides Gate Dopamine-Associated Behaviors through DRD2-Expressing Neurons. *Cell Metab* 31, 773-790 e711 (2020).

10. Throughout the manuscript there is no reference on whether the results have been replicated in independent experiments.

Response:

All the experiments were independently repeated 2 - 3 times. We have now included this information for each experiment in the Figure legends.

Reviewer #3:

Women are more likely to experience major depressive disorder (MDD) and severe symptoms, however, underlying biological mechanisms that could explain these sex differences are still unclear. In recent years, neuroimmune mechanisms of depression have received increasing attention. Nevertheless, the potential role of the meningeal lymphatic system, which is essential to drain excess macromolecules from the brain parenchyma, has yet to be explored. Here, the authors took advantage of the subchronic variable stress (SCVS) mouse model of depression to investigate efficiency of brain drainage function following stress exposure in a sex-specific manner. Indeed, the SCVS paradigm induces depression-like behaviors in female but not male mice allowing characterization of sex differences. To confirm that stress-induced alterations in meningeal lymphatics structure and function are involved in the establishment of anxiety- and depression-like behaviors, experiments were performed using a viral-mediated strategy to overexpress vascular endothelial growth factor C (vegfc, gain-of-function) or vegf receptor 3 mutants (loss-of-function). These manipulations affect activation of dopaminergic neurons in the ventral tegmental area (VTA) and astrocytes in the medial prefrontal cortex (mPFC). Finally, RNA-seq analysis were performed on sorted meningeal lymphatic endothelial cells revealing altered transcriptomic profiles after SCVS exposure in female mice.

Overall, the question addressed in this study is intriguing, findings are novel, and the manuscript is clear and well written. Figures are of high quality and very informative. Nevertheless, I do have a number of concerns that need to be addressed, as listed below, as I do believe that the results presented do not accurately and convincingly support the general conclusion that meningeal lymphatics are involved in stress susceptibility or depression in a sex-specific manner.

Response:

We thank the reviewer for the positive comments and valuable suggestions. We have obtained additional experimental data and revised the manuscripts to address the concerns raised by the reviewer.

Major:

1. All experiments were performed in mice only, but the title and last sentence of the abstract are referring to human depression. This should be modified to better reflect the findings.

Response:

To better reflect the fact that the study was performed in mice, we have modified the title to "A functional role of meningeal lymphatics in sex difference of stress susceptibility in mice", and the last sentence of the abstract to "Together, our findings suggest meningeal lymphatic impairment as an aggravating factor for

promoting susceptibility to stress in mice, and that restoration of the meningeal lymphatics might be of therapeutic potential for depression.” (main text, page 3, line 55-57).

2. Chronic stress is the main environmental factor to develop human depression. Although widely used to study sex differences in rodent with regard to stress responses, the SCVS paradigm is not considered a model of chronic stress. As such, the authors should be careful when extrapolating SCVS results to human depression. Moreover, findings described may not reflect the state of meningeal lymphatics after chronic stress exposure or at later time points after SCVS. These limitations should be highlighted in the Discussion.

Response:

We agree with the reviewer and indeed, although the sub-chronic variable stress (SCVS) paradigm is a commonly used model for studying sex differences in stress-induced depressive-like behaviors in rodents, our findings should be interpreted carefully since the 6 days stress paradigm may not well-translate into the nature of human depression and other types of animal models for depression that are induced by prolonged chronic stress. We are also aware of the limitation of our current findings, which only showed the impairment of meningeal lymphatics in the SCVS female mice that was detected at 24 h and 5 days after the completion of SCVS paradigm. Whether the state of meningeal lymphatic may continue at a later time point of SCVS female mice and its mechanistic link with depressive-like behaviors in other chronic stress models remains to be determined. We have added these descriptions in the Discussion section of the revised manuscript (main text, page 21, line 486-495).

3. Characterization of LYVE1 protein level, diameters of LYVE1-labelled meningeal lymphatic vessels, coverage, and level of injected tracers in the SS, TS and COS should be performed in unstressed and stressed males to confirm sex-specific changes in brain drainage function. Lack of change in mRNA expression do not always correlate with unaltered protein structure or function. Current data do not support that SCVS induced structural and functional impairment of the meningeal lymphatics in female, but not male mice as stated in the last sentence of this section.

Response:

We have included additional experimental data to validate the lack of impairment in meningeal lymphatics (mLV) of male mice after sub-chronic variable stress (SCVS). Using immunofluorescence staining and intracisternal tracer injection, we showed that the intensity of immunofluorescence staining for LYVE1, the coverage area of LYVE1-labelled mLV as well as the diameters of mLV in the superior sagittal sinus (SSS), the transverse sinus (TS) and the confluence of sinuses (COS) of the dura mater remained unchanged after SCVS in male mice. In line with the lack of structural deficits of meningeal lymphatics, the amount of intracisternally-injected 70kD Dextran-Texas Red tracer detected in SSS, TS and COS areas of the dura mater as well as in the deep cervical lymph nodes (dCLNs) were unaltered by SCVS in male mice. These data are included in the new Figure 1 (e, k-o, r-s).

4. In Figure 2, eGFP+SCVS and VEGFC+SCVS are compared. Were unstressed mice injected with the eGFP and VEGFC viruses as well? Assessment of baseline biological and behavioral effects of VEGFC overexpression would strengthen the manuscript.

Response:

We have included additional experimental data, which showed that intracisternal injection of AAV-VEGFC to non-stressed female mice promoted meningeal lymphatic growth, as evident by increased intensity of LYVE1 immunofluorescence staining, the coverage area of LYVE1-labelled mLV as well as the diameters of mLV in the SSS, TS and COS of the dura mater. However, we found that the VEGFC treatment alone in non-stressed female mice did not affect animals' performance in the splash test, the forced swim test, the novelty suppressed feeding test and the open field test. Furthermore, the VEGFC treatment alone did not alter astrocytic protein expression (including S100 β and GFAP) in the medial prefrontal cortex (mPFC) and cFOS expression in the TH-positive dopaminergic neurons of ventral tegmental area (VTA) (the **new Supplementary Figure 5**). These new data, together with the data presented in Figure 2, suggest that enhancement of meningeal lymphatics *per se* was not sufficient to alter depression- or anxiety-like behaviors in female mice, nor did it alter the mPFC astrocytes and the VTA dopaminergic neuronal activity, but rather, increased their resilience to stress.

5. Behavioral assays after stress exposure or viral-mediated manipulations do not cover all symptomatology observed in individuals with MDD. Anhedonia for example is generally assessed using sucrose preference and social avoidance with a social interaction test. Addition of these features would bonify the present study.

Response:

We have performed additional experiments to examine whether SCVS and AAV-mediated manipulation of meningeal lymphatics altered animal behaviors in the social interaction test and the sucrose preference test. The data showed that after SCVS, female mice showed a significant decreased preference to an unfamiliar mouse over a novel object in the social interaction test, as well as a decreased preference for sucrose water, which however could be prevented by intracisternal injection of AAV-VEGFC that improved meningeal lymphatics. Furthermore, SCVS did not induce an impairment in social interaction and sucrose preference in male mice. However, intracisternal injection of AAV overexpressing VEGFR_{d1-4}, which impaired meningeal lymphatics, rendered male mice to show impaired social interaction after SCVS. However, sucrose preference in male mice was not affected by the VEGFR_{d1-4} treatment, suggesting that impairment of meningeal lymphatics is not sufficient for SCVS-induced anhedonia in male mice. Overall, these newly added data, shown in **the new Supplementary Figure 6** further support the notion that SCVS-induced changes in meningeal lymphatics contribute to sex-difference in susceptibility to depression.

6. Similarly, lack of behavioral effects when viral-mediated manipulations to overexpress vegfc are performed in male mice subjected to SCVS would greatly strengthen the sex-specific conclusions.

Response:

We have performed this experiment as suggested by the reviewer. We found that intracisternal injection of AAV overexpressing VEGFC in male mice subjected to SCVS did not affect animals' performances in the splash test, the forced swim test, the novelty suppressed feeding test and the open field test, as well as changes in body weight (the **new Supplementary Figure 10i-q**). These data suggest that there might be a "ceiling effect" for the role of meningeal lymphatics in regulating depression- and anxiety-like behaviors. For male mice in which no damage to meningeal lymphatics were observed after SCVS, enhancing meningeal lymphatics did not further reduce their depression-like and anxiety-like behaviors.

7. Uneven distribution of intracisternally-injected tracer with enrichment in the mPFC and VTA is very intriguing. Unfortunately, it is unclear right now if experiments were performed in a naïve female mouse and if yes, is exposure to SCVS altering distribution? Is this uneven distribution sex-specific?

Response:

The presentative images on the distribution of intracisternally-injected ovalbumin-Alexa Fluor 647 (OVA) tracer in brain parenchyma provided in the original Figure 3a were from naïve female mice. We have now extended this study by adding the following experiments and analyses:

(1) We have added the quantification of the OVA tracer across multiple brain regions at different time points (15 min, 30 min and 60 min) post-intracisternal injection in the naïve female mice. As shown in the representative images and the quantification graph, the distribution of intracisternally-injected tracer was uneven in the brain parenchyma, with an enrichment in the mPFC at 15-30 min post-injection (also in the periaqueductal gray at 15 min post-injection), and in the hypothalamus and the VTA at 30-60 min post-injection (**the revised Figure 3a-b**).

(2) We have performed experiments to examine whether the distribution of intracisternally-injected OVA tracer in the brain parenchyma is affected by SCVS and intracisternal-injection of AAV-VEGFC to female mice. For this experiment, we chose the 1 h post-injection time point, the same time point at which we found reduction of the tracer arrived at dura mater and dCLNs of female mice after SCVS (Figure 1). After SCVS, significantly less tracer was found in the brain parenchyma, which however was prevented by the VEGFC treatment that improved meningeal lymphatics (the **new Supplementary Figure 7a-c**). These findings are consistent with previous reports that alteration in meningeal lymphatics could lead to changes in the rate of CSF influx and ISF diffusion in the brain parenchyma (Louveau et al. 2017; Da Mesquita et al. 2018). Therefore, our data suggest that CSF influx and the ISF diffusion in female mice is impaired by SCVS, which however can be prevented by improvement of meningeal lymphatics.

(3) We have also examined the distribution of OVA tracer at 15 min, 30 min and 60 min post-intracisternal injection in the naïve male mice. As shown in the representative images and the quantification graph in the **new Figure 5a-b**, the distribution of intracisternally-injected tracer was also uneven in the brain parenchyma of male mice. Similar to female mice (the revised Figure 3a-b), there was an enrichment of tracer in the mPFC at 15-30 min post-injection, and in the hypothalamus and the VTA at 30-60 min post-injection.

References

1. Louveau A, Plog BA, Antila S, Alitalo K, Nedergaard M, Kipnis J. Understanding the functions and relationships of the glymphatic system and meningeal lymphatics. *J Clin Invest* 127, 3210-3219 (2017).

2. Da Mesquita S, et al. Functional aspects of meningeal lymphatics in ageing and Alzheimer's disease. Nature 560, 185-191 (2018).

8. Panels of Fig.3d-e could be grouped and statistically analyzed with b-c to confirm that viral-mediated manipulation can reverse completely astrocytic changes. Same comment for h-if-g. At the moment it is hard to make conclusions with normalization performed on separate graphs and thus, either unstressed controls or mice injected with only eGFP (different y-axis).

Response:

The previous naïve vs. SCVS groups and the eGFP+SCVS vs. VEGFC+SCVS were conducted on separate cohorts of animals. The samples were collected on different dates, and the immunofluorescence staining were also performed separately. It is not recommended to directly compare the immunofluorescence staining that were not performed in parallel. Therefore, we cannot directly group the original datasets for statistical analysis.

To address the reviewer's concern, we have performed additional experiments to compare three groups of female mice in parallel: (1) mice intracisternally injected with AAV overexpressing eGFP, non-stressed; (2) mice intracisternally injected with AAV overexpressing eGFP, subjected to SCVS; (3) mice intracisternally injected with AAV overexpressing VEGFC, subjected to SCVS. As shown in the **revised Figure 3c-j**, SCVS significantly reduced the expression of astrocytic proteins, S100 β and GFAP, as well as the density of S100 β + astrocytes in the mPFC of female mice, which were prevented by the VEGFC treatment that improved meningeal lymphatics. Moreover, SCVS reduced the expression of c-FOS in the tyrosine hydroxylase-positive (TH+) dopaminergic neurons in the VTA of female mice, which was also prevented by the VEGFC treatment. Collectively, these data suggest that enhancement of meningeal lymphatics by VEGFC prevented SCVS-induced abnormalities in the mPFC astrocytes and the VTA dopaminergic neurons of female mice.

9. Data provided to suggest that viral-mediated manipulation impairing meningeal lymphatic structural and function in male mice can induce development of a depression-like phenotype do not support this conclusion. Indeed, when comparing behavioral data provided in Supp.Fig.2, behavioral effects appear to be driven by the control virus with values similar for mice injected with VEGFF3d1-4+SCVS (Fig.4k-l) and unstressed controls in the ST (about 100 sec) and FST (125-130 sec). Additional tests are needed.

Response:

We respectfully disagree with the reviewer on this point. It is commonly observed that behavioral data vary from experiments to experiments, which could be influenced by subtle changes in the testing environments and differences in the batches of animals. The behavioral tests of naïve vs. SCVS groups and the behavioral tests of eGFP+SCVS vs. VEGFR3_{d1-4}+SCVS groups were conducted on separate cohorts of animals, as well as on different dates. Therefore, we can only compare the behavioral results between the two groups tested in parallel, but it is unusual and not recommended to directly compare the behavioral data among different cohorts that are tested on different dates and with different experimental manipulations.

To address the reviewer's concerns, we have performed additional experiments by comparing three groups of male mice in parallel: (1) mice intracisternally injected with AAV overexpressing VEGFR3_{d4-7}, non-

stressed; (2) mice intracisternally injected with AAV overexpressing VEGFR3_{d4-7}, subjected to SCVS; (3) mice intracisternally injected with AAV overexpressing VEGFR3_{d1-4}, subjected to SCVS. Of note, we have replaced eGFP with a more appropriate control for VEGFR3_{d1-4}, which is VEGFR3_{d4-7} derived from VEGFR3 extracellular portion but does not bind to VEGFC, as pointed out by the reviewer 2. The mice were tested in the social interaction test and the sucrose preference test. The data revealed that impairment of meningeal lymphatics by intracisternal injection of AAV overexpressing VEGFR_{d1-4} rendered male mice to show significantly and dramatically impaired social interaction after SCVS, supporting our conclusion that impairment of meningeal lymphatics rendered male mice more susceptible to SCVS. Of note, VEGFR_{d1-4} treatment did not affect sucrose preference of male mice, indicating that there are heterogenous mechanisms underlying different depression-like phenotypes in mice (the **new Supplementary Fig. 6f-j**).

10. How can modest behavioral effects observed in male mice following viral-mediated manipulation be reconciled with drastic changes in the male mPFC astrocyte morphology and VTA neuronal activation? Is it possible that distribution of an intracisternally-injected tracer would be different in males and thus, targeted regions are not appropriate here?

Response:

As raised by the reviewer, despite drastic changes in the male mPFC astrocytes morphology and cell density, and significant reduction of c-FOS expression in the tyrosine hydroxylase-positive (TH+) dopaminergic neurons in the VTA, the overexpression of VEGFR3_{d1-4} by AAV selectively induced depression-like phenotypes in the splash test, the forced swim test and social interaction of male mice subjected to SCVS (**Figure 4k, l, Supplementary Figure 6f-i**), while no significant changes were detected in other depression- and anxiety-like behaviors including novelty-suppressed feeding test, open field test and sucrose preference test (**Figure 4m-r, Supplementary Figure 6j**). Of note, similar effects were observed after ligation of afferent vessels to the dCLNs, which blocks drainage from meningeal lymphatics to the dCLNs (**Figure 6**).

We reason that there may be two factors underlying the phenotypic inconsistency between the detected changes in the brain regions and the behavioral outcomes: (1) More prolonged or larger magnitude of changes in the mPFC astrocytes and VTA dopaminergic neuronal activity may be required to generate robust depression-like behaviors in male mice (Banasz & Duman, 2008; Tye et al. 2013). (2) Our results showed that although viral-mediate VEGFR3_{d1-4} overexpression increased the susceptibility of male mice to SCVS, the VEGFR3_{d1-4} treatment alone was not sufficient to induce depression-like phenotypes (**Supplementary Figure 10a-d**). These findings indicate the requirement of stressors in addition to VEGFR3_{d1-4}-mediated meningeal lymphatic changes to induce depression-like behaviors in male mice. Of note, although 6 days of SCVS can induce depression- and anxiety-like phenotypes in female mice, to generate robust behavioral changes in male mice generally needs 21 days of chronic variable stress (Labonte et al. 2017). Therefore, it is possible that longer administration of variable stress is required for the VEGFR3_{d1-4} treatment to generate more behavioral outcomes in male mice. Nevertheless, our data indicate that there are heterogenous mechanisms underlying different depression-like phenotypes in mice.

For examining possible sex-difference in tracer distribution in the brain parenchyma, we have examined the distribution of OVA tracer at 15 min, 30 min and 60 min post-intracisternal injection in the naïve male mice. As shown in the representative images and the quantification graph in the **new Figure 5a-b**, the distribution of intracisternally-injected tracer was also uneven in the brain parenchyma of male mice. Similar to female

mice (the revised Figure 3a-b), there was an enrichment of tracer in the mPFC at 15-30 min post-injection, and in the hypothalamus and the VTA at 30-60 min post-injection. We have included the new data and the discussion in the revised version of the manuscript (**main text, page 24-25, line 562-576**).

References

1. Banasr M, Duman RS. Glial loss in the prefrontal cortex is sufficient to induce depressive-like behaviors. *Biol Psychiatry* 64, 863-870 (2008).
2. Tye KM, et al. Dopamine neurons modulate neural encoding and expression of depression-related behaviour. *Nature* 493, 537-541 (2013).
3. Labonte B, et al. Sex-specific transcriptional signatures in human depression. *Nat Med* 23, 1102-1111 (2017).

11. Transcriptomic profiling of meningeal lymphatic endothelial cells of female mice exposed or not to SCVS is highly relevant. The comparison with DEGs identified in a study related to aging are intriguing but why not explore gene lists associated with human depression? Even in the context of heterogeneous preparation confirmation of gene expression changes in human tissue databases, and identified here in stressed female mice, would be more appropriate.

Response:

Comparison between DEGs identified in our study with those from human studies is a great suggestion. We have checked the published database, but all the accessible transcriptomic studies on human depression were conducted on brain parenchyma, but not on meningeal lymphatic endothelial cells (LECs). The cell type composition of brain parenchyma is quite distinct from LECs, rendering hard to directly compare the changes in transcriptomics profiles of the two. Furthermore, previous studies have noticed that, even with brain parenchyma, changes in transcriptomic profiles associated with depression phenotypes in human are highly brain region-specific (Labonté et al. 2017; Bagot et al. 2016). Therefore, in the previous version of the manuscript, we did not compare the DEGs identified in the LECs with gene lists associated with human depression. Instead, we compared with those identified in the LECs in an aging study, because both aging and stress impair meningeal lymphatics (at least in the animal models), though occurring over different timescales. We feel that comparison of gene expression changes in the LECs induced by SCVS versus aging will help identify key common mechanisms that may contribute to the impairment of meningeal lymphatics by different physiological processes.

To address the reviewer's point on whether gene expression changes identified in the stressed female mice in the current study may share any similarity with those associated with human depression, we have added additional comparison of the DEGs identified in the current study with those identified in a previous study by Labonté et al. (2017), which explored changes in transcriptional profiles across 6 brain regions from postmortem tissues of female patients suffered from major depressive disorder (MDD). This analysis identified collectively 12 up-regulated and 9 down-regulated DEGs shared by LECs of female mice subjected to SCVS with at least one of the examined brain regions by Labonté et al. Among these DEGs, the most noticeable one is *Trpm1*, which encodes a transient receptor potential cation channel. *Trpm1* was found to be upregulated in the LEC of female mice subjected to SCVS in our study as well as in ventral subiculum, dorsolateral prefrontal

cortex, nucleus accumbens and anterior insula of female MDD patients, indicating it as a candidate gene worthy of further exploration for the development of depression. We also noticed that Col1a2, a down regulated gene in the LEC shared by aging and stress, was also downregulated in the nucleus accumbens of female MDD patients. These new findings are added to the **revised Figure 7I**.

References

1. Labonte B, et al. Sex-specific transcriptional signatures in human depression. Nat Med 23, 1102-1111 (2017).
2. Bagot RC, et al. Circuit-wide Transcriptional Profiling Reveals Brain Region-Specific Gene Networks Regulating Depression Susceptibility. Neuron 90, 969-983 (2016).

Minor:

1. It is unclear in the abstract which brain regions were characterized for brain drainage function. Same comment for injections of the AAVs. VTA and mPFC are mentioned at the end but it appears disconnected.

Response:

Unfortunately, due to the word limitation of the abstract (which cannot exceed 150 words), we have to further shorten the abstract from the original version, and therefore cannot include the details of our research design in the abstract.

Regarding the questions raised by the reviewer: (1) as mentioned in the Introduction section, the meningeal lymphatics constantly drain fluid and macromolecules from the CNS to the periphery. Therefore, the impairment of female meningeal lymphatics by SCVS likely affect the drainage function of the whole brain. Indeed, in a newly added experiment, we found significantly less tracer in the overall brain parenchyma after SCVS, which however was prevented by the VEGFC treatment that improved meningeal lymphatics (the **new Supplementary Figure 7a-c**). These data suggest that CSF influx and the ISF diffusion in female mice is impaired by SCVS, which can be prevented by improvement of meningeal lymphatics.

(2) The AAV expressing VEGFC, VEGFR3 mutants or eGFP was injected into the cisterna magna. We and others have found that after injection by this method, the AAV infected cells were mostly localized in the transverse sinus (TS), the confluence of sinuses (COS) and part of the superior sagittal sinus (SSS) that covering the olfactory bulb, and near LYVE1-labelled meningeal lymphatic vessels. We did not observe infected cells in the brain parenchyma, except occasionally few cells in the cerebellum that near the injection site. Of note, similar restricted distribution of intracisternally injected AAV-VEGFC was also noted in a previous study, which reported that this viral approach did not affect meningeal blood vessel coverage or proliferation of neural stem cells in the hippocampus, which could potentially be mediated by VEGFC signaling (Da Mesquita et al. 2018).

(3) The mPFC and the VTA are involved in the brain circuit critical for the pathological development of depression (Tye et al. 2013; Maier et al. 2010). Furthermore, in our study, we found an enrichment of intracisternally-injected tracer in the mPFC at 15-30 min post-injection, and in the VTA at 30-60 min post-injection, indicating that the mPFC and the VTA might be the active spots for CSF-ISF exchange (the **revised Figure 3a-b** and the new **Figure 5a-b**). Therefore, we chose to focus on the mPFC and the VTA to examine how impairment of meningeal lymphatics by stress might affect brain regions engaged in emotional regulation.

References

1. Da Mesquita S, et al. Functional aspects of meningeal lymphatics in ageing and Alzheimer's disease. *Nature* 560, 185-191 (2018).
2. Tye KM, et al. Dopamine neurons modulate neural encoding and expression of depression-related behaviour. *Nature* 493, 537-541 (2013).
3. Maier SF, Watkins LR. Role of the medial prefrontal cortex in coping and resilience. *Brain Res* 1355, 52-60 (2010).

2. Timeline of SCVS vs tissue collection should be justified. Why wait until day 11? This is unconventional with most studies collecting tissue the day after the last stress exposure session vs 72h here. Were other time points (24h, 48h) analyzed for Lyve1 expression with no difference observed?

Response:

In the literature, tissue collection was usually carried out at 24-72 h after the final behavioral test, to avoid potential complication of acute stress from behavioral testing on animals (e.g. Farris et al. 2020; Weger et al. 2020; Dearing et al. 2021). In our laboratory, we usually collect tissues at 48-72 h after behavioral tests.

For this study, we did examine the changes in meningeal lymphatics of female mice at 24 h after the last stress episode of SCVS, which revealed significant reduction in the coverage area of meningeal lymphatic vessels and the diameters of LYVE1-labelled meningeal lymphatic vessels, as well as a trend towards decreased intensity of LYVE1 immunofluorescence staining. These data suggested that the impairment of meningeal lymphatics by SCVS in female mice could be observed as early as 24 h post-SCVS. These data are included in the **new Supplementary Figure 4**.

For most of the study, we chose the 72 h post-testing as the time point for tissue collection (5 days after the last stress episode), because we hope to explore relatively long-lasting changes in the meningeal lymphatics by SCVS. We have now included the justification for this timeline in the Methods section (**main text, page 29, line 674-680**).

References

1. Farris SP, Tiwari GR, Ponomareva O, Lopez MF, Mayfield RD, Becker HC. Transcriptome Analysis of Alcohol Drinking in Non-Dependent and Dependent Mice Following Repeated Cycles of Forced Swim Stress Exposure. *Brain Sci* 10(5): 275 (2020).
2. Weger M, et al. Mitochondrial gene signature in the prefrontal cortex for differential susceptibility to chronic stress. *Sci Rep* 10(1): 18308 (2020).
3. Dearing C, et al. Glucoregulation and coping behavior after chronic stress in rats: Sex differences across the lifespan. *Horm Behav* 136: 105060 (2021).

3. Mouse cohorts that went through behavioral characterization (Supp. Fig.1-2) are twice larger for female vs male mice, but it is unclear why. It is surprising that power analysis gave n=20-25 for females vs only n=9 for males. Comparable number of animals for each sex may render SCVS-induced behavioral effects in males significant.

Response:

We used different amounts of female and male mice in the original Suppl. Fig.1-2 because the mice were not only used for behavioral characterization, but also their tissues were collected for biochemical characterizations, including (1) qPCR analysis of Lyve1 expression in the dura mater for male and female mice, and (2) analyses of meningeal lymphatics by immunofluorescence staining and the drainage of intracisternally-injected tracer, which were previously performed only on female mice. Therefore, while combining behavioral data from all the experiments, the animal number for female mice was twice larger than the male mice. We would like to emphasize that in each independent cohort of the replicated experiments (each with n = 4-6 animals per group), we consistently observed that SCVS induced depressive- and anxiety-like phenotypes in female mice, but not in male mice.

To address the reviewer's concern, we have **revised the Supplementary Figure 1**, which includes the behavioral data from the first two repeats of experiments, in which similar number of female mice were used as male mice used for the behavioral analysis in the Supplementary Figure 2. As shown in this revised Supplementary Figure 1, SCVS induced significant weight loss, decreased grooming in the splash test, increased immobility in the forced swim test, increased latency to feeding in the novelty-suppressed feeding test and decreased exploration of the center zone of an open field arena. These data suggest that SCVS induced both depression- and anxiety-like behaviors in female mice.

4. In the Reporting Summary, “No data were excluded” is mentioned in Life sciences study design, however, the number of mice for behavioral tests differs between graphs (ex: Supplementary Fig 1 and 2, Fig 1, Fig 2, Fig 4). Please explain in detail how outliers were determined.

Response:

We thank the reviewer for raising the question in the difference of animal numbers between graphs in certain behavioral results. After checking the experimental datasets, we realized that this difference was occurred due to the removal of certain outliers but the associated data points in other behavioral graphs were remained, resulting in the inconsistency of animal numbers between graphs as mentioned by the reviewer. For determining outliers in the behavioral datasets, Shapiro-Wilk normality test was firstly used. In the datasets which were normally distributed, outliers were identified as being greater than 2 standard deviations from the mean and excluded from statistical analysis as described previously by Dion-Albert L et al (2022). We have added the detailed method of outlier determination to the Methods section of the revised manuscript (**main text, page 39, line 909-912**) and the Reporting Summary.

References

1. Dion-Albert L, et al. Vascular and blood-brain barrier-related changes underlie stress responses and resilience in female mice and depression in human tissue. *Nat Commun* 13, 164 (2022).

5. Were proper statistic tests conducted to assume normality of the datasets (Kolmogorov–Smirnov, Shapiro–Wilk test, etc.)?

Response:

We have conducted the Shapiro-Wilk test to evaluate the normality of the datasets. Based on the results from the test, most of the datasets were normally distributed. We used the Mann-Whitney test for two-group comparisons of non-normally distributed datasets. We have now included this information to the revised Methods section (**main text, page 39, line 909-914**) and corresponding Figure Legends.

6. The Data availability statement is insufficient and no access to gene lists produced by RNA sequencing in suitable repositories was provided.

Response:

The RNA sequencing data generated for this study can be found in the GEO repository under accession number GSE201368 (<https://www.ncbi.nlm.nih.gov/geo/query/acc.cgi?acc=GSE201368>). We have added this information to the Data availability section (**main text, page 39, line 919-921**).

REVIEWER COMMENTS

Reviewer #1 (Remarks to the Author):

The authors have done an outstanding job of addressing my comments. Congrats on this exciting work. - John Lukens

Reviewer #2 (Remarks to the Author):

The authors did a very good job at addressing all of my concerns with new experiments and a thorough revision of the main text and methods. The methods are described with more detail. I would still encourage the authors to include even more information in the methods' sections to improve the rigor of the manuscript and allow reproducibility of the findings.

The new body of experimental evidence provides a more clear idea of the biology underlying the described phenomena. I have only one additional recommendation that should be easy to address: - The new data suggest that CCL6 is underlying the alterations in meningeal lymphatic function and behavior in female subjected to SCVS. Since the authors have generated RNAseq data from dural LECs, it would be important to check the levels of normalized mRNA reads for Ccr1 (the gene encoding the receptor for CCL6) in LECs from the different groups. This will allow us to understand: 1) if Ccr1 is at all expressed by meningeal LECs, and if we are witnessing a potential autocrine and cell-autonomous response; 2) if SCVS alters the expression of Ccr1 in LECs; and 3) if CCL6 is acting in cell non-autonomous manner, by signaling on dural cellular intermediaries other than LECs, which will then underly the loss of meningeal lymphatic coverage.

Except for this last data request, I have nothing else to add. This is a very interesting study.

Reviewer #3 (Remarks to the Author):

The authors addressed all my concerns and suggestions and thus, I can now recommend publication.

Response to reviewers

Manuscript: NCOMMS-21-44233B

Dai et al. A functional role of meningeal lymphatics in sex difference of stress susceptibility in mice

We are pleased to submit our revised manuscript entitled “A functional role of meningeal lymphatics in sex difference of stress susceptibility in mice” for consideration for publication in *Nature Communications*.

We thank all the reviewers for their valuable critiques and suggestions during the review process, which make our manuscript much stronger. We are also grateful for their thorough reviews of our revised manuscript. Please find below our point-by-point response to the comments and suggestions from the reviewers.

Reviewer #1 (Remarks to the Author):

The authors have done an outstanding job of addressing my comments. Congrats on this exciting work. - John Lukens

Response:

We thank Dr. Lukens for the very positive comment on our manuscript.

Reviewer #2 (Remarks to the Author):

The authors did a very good job at addressing all of my concerns with new experiments and a thorough revision of the main text and methods. The methods are described with more detail. I would still encourage the authors to include even more information in the methods' sections to improve the rigor of the manuscript and allow reproducibility of the findings.

The new body of experimental evidence provides a more clear idea of the biology underlying the described phenomena. I have only one additional recommendation that should be easy to address: - The new data suggest that CCL6 is underlying the alterations in meningeal lymphatic function and behavior in female subjected to SCVS. Since the authors have generated RNAseq data from dural LECs, it would be important to check the levels of normalized mRNA reads for Ccr1 (the gene encoding the receptor for CCL6) in LECs from the different groups. This will allow us to understand: 1) if Ccr1 is at all expressed by meningeal LECs, and if we are witnessing a potential autocrine and cell-autonomous response; 2) if SCVS alters the expression of Ccr1 in LECs; and 3) if CCL6 is acting in cell non-autonomous manner, by signaling on dural cellular intermediaries other than LECs, which will then underly the loss of meningeal lymphatic coverage.

Except for this last data request, I have nothing else to add. This is a very interesting study.

Response:

We thank the reviewer for the positive comments and thorough reviews of our revised manuscript. Following are the responses to the additional suggestions of the reviewers:

(1) Regarding the point to include more information in the Methods section, we have revised the whole Methods section to include additional experimental details. The newly added information is highlighted by blue text in the revised manuscript (main text, page 26 – 41).

(2) Regarding the point on CCR1, we examined the normalized counts of *Ccr1* in our RNA-seq data of meningeal lymphatic endothelial cells (LECs) and compared non-stressed naïve female mice with female mice experienced sub-chronic variable stress (SCVS). The data showed that the expression of *Ccr1* was low in the meningeal LECs of naïve female mice (average counts per million reads: 0.662). After SCVS, *Ccr1* expression was elevated in 2 out of the 5 biological replicates, but remained low in the other 3 replicates (average CPM for the SCVS group: 19.660), and was not significantly different from the naïve group (adjusted $p = 0.999$). Based on these data, we suspect that CCL6 is likely acting in a cell non-autonomous manner by signaling on dural cellular intermediaries other than LECs, which in turn leads to reduction in meningeal lymphatic coverage of female mice after SCVS. A description of *Ccr1* expression data is included in the revised manuscript, highlighted by blue text (main text, page 19-20, line 448-456).

Reviewer #3 (Remarks to the Author):

The authors addressed all my concerns and suggestions and thus, I can now recommend publication.

Response:

We thank the reviewer for the recommendation of our manuscript for publication.